# Robust Vision-Language Models via Manifold-Adversarial Adapters

Hao Li [1]   Zeyu Xiao [2]   Junhao Zhou [1]   Peng Liu [1]   Yang Zhao [1]   Wei Jia [1]

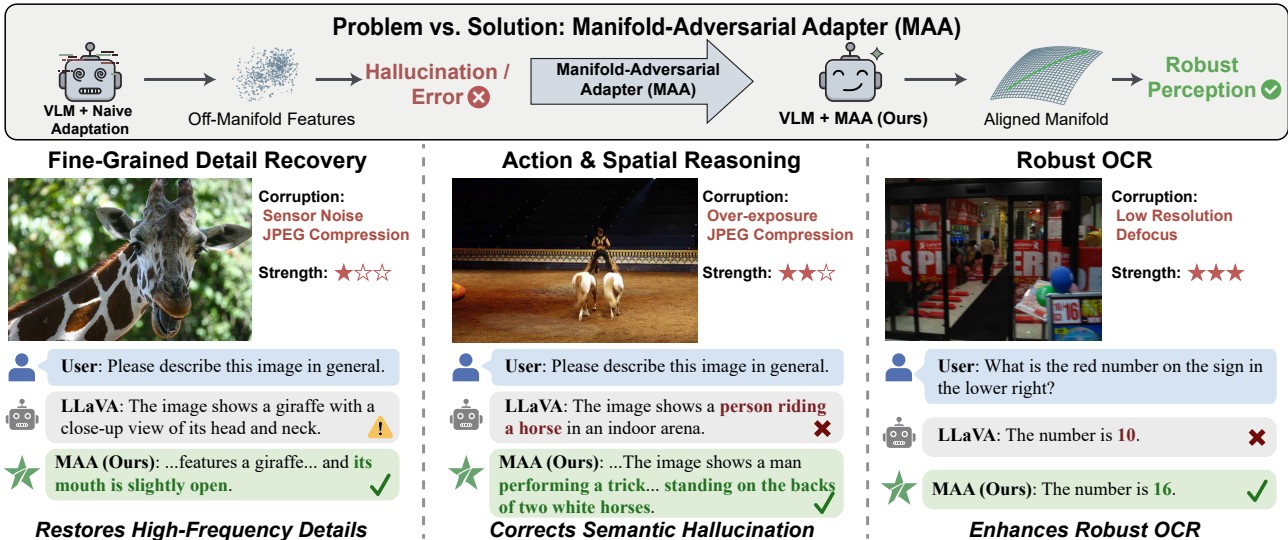

*Figure 1.* Manifold-adversarial adapters (MAA) for robust VLMs. Naive adaptation under real-world corruptions can pull visual features off the in-distribution semantic manifold, leading to hallucinations and reasoning errors. We propose MAA, layer-wise modules that correct corruption-induced feature drift by combining paired feature distillation with an adversarial manifold constraint. By steering corrected features back onto the aligned manifold, MAA improves robustness across fine-grained perception, spatial reasoning, and OCR without modifying the VLM backbones.

## Abstract

Vision-language models (VLMs) have progressed rapidly with large-scale high-quality data and adaptation strategies, yet remain brittle under real-world corruptions, where both visual recognition and language-grounded reasoning degrade. Beyond cascaded image restoration, a natural alternative is parameter-efficient adaptation that aligns corrupted features with clean references; however, Euclidean alignment alone is not semantics-preserving and can even harm downstream reasoning. We attribute this to a *semantic misalignment gap*, where features become geometrically closer while drifting off the in-distribution support on which multimodal reasoning is calibrated. To address this, we propose Manifold-Adversarial Adapters (MAA), parameter-efficient layer-wise modules for a frozen vision encoder that explicitly steer corrupted features back onto the clean in-distribution manifold rather than merely shrinking feature-space distance. MAA combines paired feature self-distillation with a token-level adversarial manifold constraint to prevent off-manifold semantic shortcuts. At inference, only the adapters are retained, enabling single-stage robustness with negligible overhead and avoiding the latency and semantic drift of restoration pipelines. Across benchmarks and corruption settings, MAA consistently improves performance over strong baselines. Code is available at here.

## 1. Introduction

Vision-language models (VLMs) have recently advanced rapidly, driven by large-scale training on high-quality, predominantly clean image–text data and careful adaptation strategies (OpenAI, 2023; Liu et al., 2023a; Yin et al., 2024; Wang et al., 2024a). In real-world deployments, however, this success does not readily extend to corrupted visual

---

[1]Hefei University of Technology, Hefei, China [2]National University of Singapore, Singapore. Correspondence to: Zeyu Xiao <zeyuxiao1997@163.com>.

*Proceedings of the 43rd International Conference on Machine Learning*, Seoul, South Korea. PMLR 306, 2026. Copyright 2026 by the author(s).

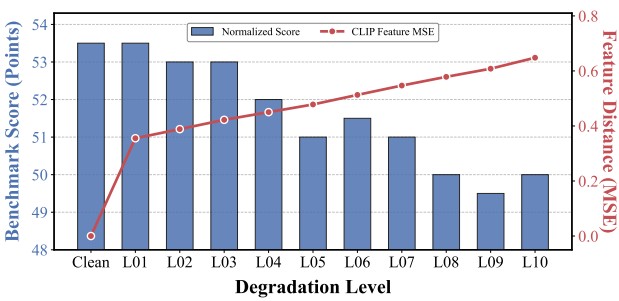

Figure 2. Severity-controlled corruption analysis. We apply synthetic corruptions with increasing severity (L01-L10) to high-quality images and measure the CLIP vision-feature MSE to the clean reference and a lightweight downstream score. As severity increases, feature MSE rises and downstream performance drops, linking feature drift to semantic degradation.

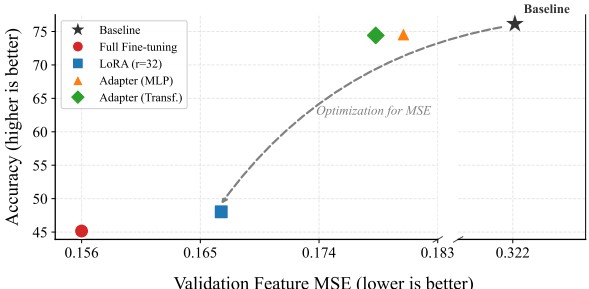

Figure 3. Geometric proximity vs. Semantic capability. Naively minimizing feature MSE during PEFT reduces the validation feature distance, but can severely hurt semantic performance on multimodal benchmarks.

inputs: performance degrades noticeably under common corruptions such as blur, noise, compression artifacts, and illumination shifts (Hendrycks & Dietterich, 2019; Verma et al., 2024). This fragility limits degradation-sensitive applications, including autonomous driving (Xie et al., 2025) and medical imaging (Liu et al., 2019). While recent benchmarks systematically characterize these failures (Li et al., 2025a; Zhang et al., 2024; Sui et al., 2025; Xinkuan et al., 2025) and prior work largely focuses on adversarial robustness (Liu et al., 2024a; Zhao et al., 2023; Schlarmann & Hein, 2023), general and principled methods for improving VLM robustness to natural corruptions remain limited.

Two common remedies exhibit clear limitations. One is to prepend an image restoration (IR) model before VLM inference; however, IR objectives (*e.g.*, PSNR and SSIM) are often misaligned with high-level semantics, and restoration artifacts can further disrupt multimodal alignment, while the resulting two-stage pipeline incurs additional latency and deployment overhead. Another is to apply parameter-efficient fine-tuning (PEFT) on corrupted data to align corrupted features with clean references, yet minimizing Euclidean distance alone is not semantics-preserving and can even degrade downstream reasoning. Together, these limitations motivate an *in-model*, feature-level robustness mechanism that corrects corruption-induced representation drift while preserving semantic structures required for VLM reasoning.

A natural starting point is that real-world corruptions induce a distribution shift in the vision encoder (Hendrycks & Dietterich, 2019), causing features to drift from their clean counterparts. If so, aligning corrupted features with clean ones may improve robustness. We validate this hypothesis in a controlled setting (Appendix C). Using high-quality Flickr2K images (Lim et al.) as clean references, we synthesize corruption severities L01-L10 by progressively strengthening a fixed degradation pipeline (noise, blur, compression, and illumination). We measure (i) the

MSE between CLIP vision-encoder features of degraded images and their clean references (Radford et al., 2021), and (ii) a lightweight downstream score. As shown in Figure 2, increasing corruption severity leads to larger feature MSE and lower downstream performance. This motivates a feature-level objective $\mathcal{L}_{\mathrm{mse}} = \|Z_{\mathrm{deg}} - Z_{\mathrm{ref}}\|_2^2$, where $Z_{\mathrm{ref}} = E(x_{\mathrm{ref}})$ is produced by a frozen vision encoder $E(\cdot)$ from a clean reference image $x_{\mathrm{ref}}$, and $Z_{\mathrm{deg}}$ is the corresponding feature from the corrupted image after inserting a learnable module.

Although LoRA (Hu et al., 2022) and vanilla adapters (Houlsby et al., 2019) trained on large-scale clean and corrupted pairs reduce feature MSE on held-out data, semantic performance on comprehensive multimodal benchmarks (*e.g.*, MMBench (Liu et al., 2024b)) degrades markedly. In extreme cases, LoRA or full fine-tuning can even catastrophically damage the pretrained vision encoder (Figure 3). This paradox shows that Euclidean proximity is not semantics-preserving. Features can move closer to clean references while becoming unreliable for downstream reasoning. We attribute this to the *semantic misalignment gap (SMG)*. Pretrained vision encoders such as CLIP (Radford et al., 2021) and SigLIP (Zhai et al., 2023) implicitly define an in-distribution feature support $\mathcal{M}$ on which multimodal reasoning is calibrated. Naive MSE minimization may push features toward clean targets along directions that leave $\mathcal{M}$, yielding effectively out-of-distribution representations.

To address SMG, we propose the Manifold-Adversarial Adapter (MAA), a parameter-efficient and adapter-based design implemented as layer-wise adapters in the vision encoder. Each MAA employs a three-branch residual structure that combines window self-attention, token-wise MLP, and max pooling to capture complementary local, channel-wise, and robust aggregation cues, enabling expressive yet conservative corrections. MAA is trained with a dual objective combining paired semantic self-distillation with an

adversarial manifold constraint, steering corrected features toward clean counterparts while keeping updates near the in-distribution support. By applying small, controlled residual corrections across layers, MAA follows the progressive evolution of pretrained semantics and suppresses Euclidean shortcut updates that destabilize reasoning. At inference, only the adapters are retained, yielding a single-stage robust VLM with negligible overhead and without reliance on two-stage restoration pipelines. Across diverse benchmarks, MAA consistently improves robustness over strong baselines.

Our contributions are threefold: (1) We show in controlled experiments that stronger corruptions induce larger feature drift and semantic degradation, and that minimizing feature MSE via PEFT can be ineffective or harmful. (2) We formalize the underlying failure as the SMG, where Euclidean alignment reduces feature distance but moves representations away from the in-distribution semantic support required for multimodal reasoning. Based on this analysis, we propose MAA, which couples self-distillation with an adversarial manifold constraint to encourage on-support corrections. (3) We build a multi-source, locality-aware corruption synthesis recipe and show that MAA achieves consistent robustness gains with modest overhead.

## 2. Related Works

**Benchmarks for Corruption Robustness.** Robustness evaluation under common corruptions is first popularized in unimodal settings by ImageNet-C (Hendrycks & Dietterich, 2019). Recent benchmarks extend this paradigm to vision-language models. MMCBench (Zhang et al., 2024) evaluates multimodal generation performance under corruptions across model checkpoints. R-Bench (Li et al., 2025a) focuses on real-world degradations to reduce the gap between synthetic and realistic corruptions, while MLLM-IC (Xinkuan et al., 2025) introduces a fine-grained taxonomy covering 40 corruption types and 34 sub-capabilities. Bench-C (Sui et al., 2025) proposes the Robustness Alignment Score to analyze structural deviations in model predictions. Overall, these benchmarks consistently show that VLMs remain vulnerable to natural corruptions, while principled methods to improve intrinsic corruption robustness are still limited.

**Image Restoration for Recognition and VLMs.** IR has progressed from single-degradation models (Zhang et al., 2017; Liang et al., 2021) to unified all-in-one frameworks (Li et al., 2022). Recent approaches incorporate semantic priors from vision-language models; for example, DA-CLIP (Luo et al., 2024b) introduces a degradation-aware controller guided by CLIP. In practice, IR systems such as GAN-based enhancement (*e.g.*, Real-ESRGAN (Wang et al.)), prompt-driven restoration (*e.g.*,

PromptIR (Potlapalli et al., 2023)), and mixture-of-experts designs (*e.g.*, MoCE-IR (Zamfir et al., 2025)) are often used as generic pre-processing modules under diverse degradations. Despite these advances, using IR as a preprocessing step for VLMs is often suboptimal. First, IR objectives (*e.g.*, PSNR and SSIM) emphasize pixel fidelity and may not improve multimodal reasoning. Second, restoration artifacts and distribution shifts can distort multimodal alignment and harm downstream performance (Xing et al., 2025). Third, two-stage pipelines introduce substantial computation and latency. These limitations motivate internal, feature-level adaptation that directly targets corruption-induced representation shift without relying on external restoration. (For extended discussion, see Appendix G.)

## 3. Method

### 3.1. Preliminaries

We study robustness adaptation for VLMs under common real-world corruptions. Given a frozen VLM with vision encoder $E(\cdot)$ (*e.g.*, a CLIP ViT in LLaVA) and language model $G(\cdot)$, we train parameter-efficient vision-encoder adapters to correct corrupted representations while preserving pretrained semantics for downstream large language model (LLM) reasoning.

We assume paired samples $(x_{\text{ref}}, x_{\text{deg}})$, where $x_{\text{deg}}$ is a corrupted version of a clean reference image $x_{\text{ref}}$ generated by a degradation pipeline (Section 3.5) or collected from the real data. During the inference stage, only $x_{\text{deg}}$ is available and the reference branch is used solely for training supervision. We denote the adapter-augmented encoder as $E_{\theta_A}$ with trainable parameters $\theta_A$, while all original VLM parameters remain frozen.

Let $H \in \mathbb{R}^{(N+1) \times d_v}$ denote the final-layer token sequence of the vision encoder. $H$ consists of one [CLS] token and $N$ patch tokens (for a ViT with patch size $p$ and input resolution $r \times r$, $N = (r/p)^2$). Following LLaVA-style architectures, visual tokens are projected to the hidden space by a frozen projector $P(\cdot)$. We compute losses on the projected tokens (excluding [CLS]) via:

$$
\begin{aligned}
Z_{\text{ref}} &= P(E(x_{\text{ref}})_{1:}) \in \mathbb{R}^{N \times d}, \\
Z_{\text{deg}} &= P(E_{\theta_A}(x_{\text{deg}})_{1:}) \in \mathbb{R}^{N \times d},
\end{aligned}
\tag{1}
$$

where $d$ is the LLM hidden size and $(\cdot)_{1:}$ denotes removing the [CLS] token. We compute losses in the projected space because these tokens are the representations consumed by the LLM, and multimodal alignment and reasoning are formed in this space during training, making it the most relevant domain for preserving downstream semantics.

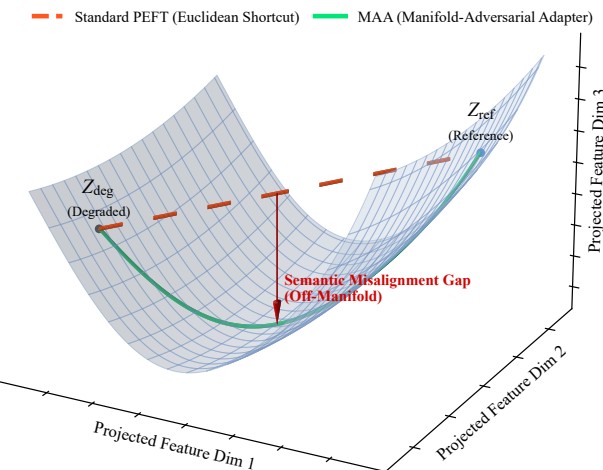

*Figure 4.* Conceptual illustration of SMG. MSE-only PEFT can reduce the Euclidean distance from the degraded feature $Z_{\text{deg}}$ to the clean reference $Z_{\text{ref}}$ via an off-manifold shortcut, deviating from the in-distribution manifold $\mathcal{M}$. MAA combines paired feature distillation with an adversarial manifold constraint, encouraging corrected features to stay on $\mathcal{M}$.

### 3.2. Semantic Misalignment Gap

A natural approach to improve robustness is to reduce the feature discrepancy between corrupted and clean images. However, our motivation study (Figure 3) reveals a counterintuitive outcome: naively minimizing feature MSE via PEFT does reduce the distance to clean references, but it substantially degrades downstream multimodal capability. This suggests that pointwise Euclidean alignment alone is insufficient to preserve semantic validity in a frozen VLM.

To quantify this failure mode, we compare corrected features to the clean feature manifold using distributional divergence and representational similarity metrics. We find that MSE-based adaptation can reduce the paired $\ell_2$ distance, yet increases distributional divergence (*e.g.*, MMD$^2$ (Gretton et al., 2012), KID (Bińkowski et al., 2018), and Energy (Rizzo & Székely, 2016)) and decreases representational similarity (*e.g.*, CKA (Kornblith et al., 2019)) to clean features. We term this mismatch the SMG.

We interpret SMG from a manifold perspective. Pretrained vision encoders implicitly induce an in-distribution feature manifold $\mathcal{M}$ on which downstream reasoning is calibrated. Minimizing feature MSE enforces only pairwise proximity to clean references and does not constrain corrected features to remain on $\mathcal{M}$. As a result, optimization can take Euclidean shortcuts through low-density, out-of-distribution regions, producing representations that are closer to $Z_{\text{ref}}$ yet less compatible with downstream reasoning. Motivated by this observation, we adopt a dual-objective training strategy that distills paired clean features to guide correction while aligning corrected features with the clean feature distribution via an adversarial discriminator to discourage off-manifold shortcuts. This analysis indicates that distri-

butional constraints are essential for preserving semantic validity under corruption, beyond minimizing feature distance alone.

### 3.3. Layer-wise Manifold-Adversarial Adapters

To mitigate SMG, we insert parameter-efficient, layer-wise adapters into each of the $L$ transformer blocks of the frozen vision encoder, enabling small residual corrections for corruption-induced drift while limiting unintended semantic shift. Let $H^\ell \in \mathbb{R}^{(N+1) \times d_v}$ denote the token features after the $\ell$-th ViT block (one `[CLS]` token and $N$ patch tokens). Note that $H^\ell$ represents the intermediate backbone features, distinct from the final projected features $Z$ used in the loss objective (Eq. 6).

The adapter at layer $\ell$, denoted as a function $A_\ell(\cdot)$, applies a residual correction:

$$\tilde{H}^\ell = H^\ell + A_\ell(H^\ell), \tag{2}$$

and the rectified feature $\tilde{H}^\ell$ is fed into the next vision block. Only the adapter parameters $\theta_A$ are trainable; the pretrained vision encoder and the projector $P$ remain frozen.

MAA targets corruption repair under a frozen backbone, where the adapter must correct corruption-induced drift without rewriting pretrained semantics that downstream reasoning relies on. Since real-world corruptions are heterogeneous and often spatially non-uniform, each MAA module uses three complementary branches: (1) window self-attention (Liu et al., 2021) to leverage local context and selectively repair spatially varying structural degradation, (2) a token-wise MLP to perform lightweight non-linear channel re-calibration and compensate feature-statistic shifts induced by noise, compression, or illumination changes, and (3) max pooling to provide a simple yet robust local aggregator that suppresses noisy outliers while preserving salient responses. These branches cover distinct correction mechanisms, reducing the chance that optimization relies on a single brittle shortcut. Finally, all branches are gated and zero-initialized to implement an identity mapping at the start of training, ensuring conservative updates and allowing the adapter to learn only the minimal corrections needed to restore on-support representations (Wang et al., 2024b).

Window attention and pooling operate on patch tokens by reshaping them into a 2D grid (the `[CLS]` token is kept unchanged for these spatial branches), while the MLP is applied to all tokens. We compute the aggregated correction $A_\ell(H^\ell)$ as:

$$\begin{aligned}
U &= \alpha_{\text{attn}} \, \text{WAttn}(H^\ell), \\
V &= \alpha_{\text{mlp}} \, \text{MLP}(H^\ell), \\
W &= \alpha_{\text{pool}} \, \text{Pool}(H^\ell), \\
A_\ell(H^\ell) &= U + V + W,
\end{aligned} \tag{3}$$

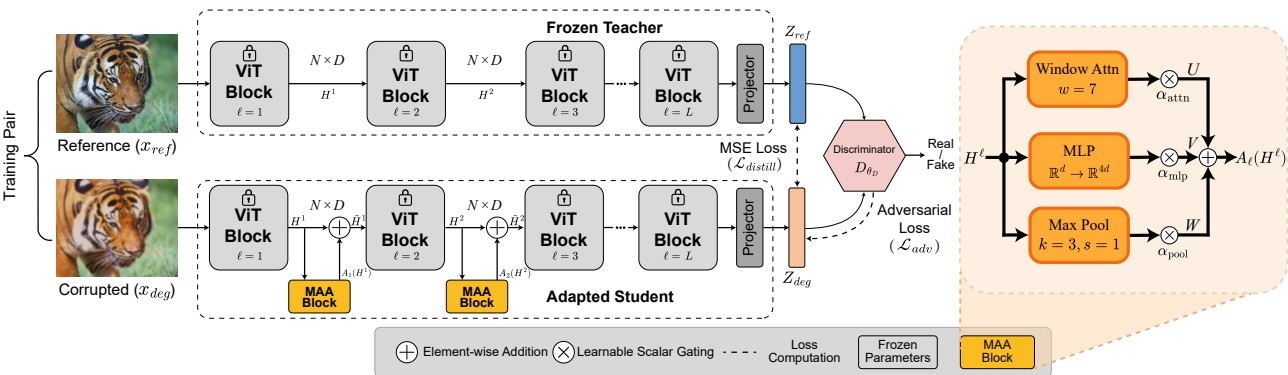

*Figure 5.* Overview of the MAA framework. The pipeline consists of a dual-stream architecture: a *Frozen Teacher* (top) processing the reference image $x_{\text{ref}}$, and an *Adapted Student* (bottom) processing the corrupted image $x_{\text{deg}}$. MAA modules (orange) are inserted into each vision transformer block of the student to rectify features. Training is guided by a paired feature distillation loss $\mathcal{L}_{distill}$ between $Z_{\text{ref}}$ and $Z_{\text{deg}}$, together with an adversarial manifold constraint $\mathcal{L}_{adv}$ implemented by a discriminator $D$ operating on projected patch tokens. The backbone vision encoder and projector are frozen; only MAA blocks (and $D$ during training) are trainable.

where $\alpha_{\text{attn}}, \alpha_{\text{mlp}}, \alpha_{\text{pool}}$ are learnable scalar gates controlling the correction magnitude of each branch.

### 3.4. Distillation and Adversarial Manifold Alignment

We train the adapters with a dual objective to explicitly address the SMG: paired feature distillation enforces token-level recovery toward clean references, while adversarial manifold alignment constrains the corrected features to remain on the clean in-distribution manifold in the projected token space. For clarity, we denote the adapted student features from corrupted inputs as $Z_{\text{corr}}$, which is equivalent to $Z_{\text{deg}}$, *i.e.*, $Z_{\text{corr}} \equiv Z_{\text{deg}}$.

We first impose a pairwise semantic constraint to guide corrupted features toward their clean counterparts. Teacher features are extracted by a pristine frozen vision tower $E(\cdot)$ from the clean reference image, while student features are produced by the adapter-augmented tower $E_{\theta_A}(\cdot)$ from the corrupted image. Let $Z_{\text{ref}} = P(E(x_{\text{ref}})_{1:})$ and $Z_{\text{corr}} = P(E_{\theta_A}(x_{\text{deg}})_{1:})$ denote the projected patch tokens (excluding $\texttt{[CLS]}$). We minimize the token-wise mean squared error:

$$\mathcal{L}_{\text{distill}} = \frac{1}{N} \sum_{i=1}^{N} \left\| Z_{\text{corr}}^{(i)} - Z_{\text{ref}}^{(i)} \right\|_2^2, \tag{4}$$

which enforces pointwise semantic consistency but alone does not prevent off-manifold corrections.

To complement the pairwise constraint with a distribution-level regularization, we introduce a token-wise discriminator $D_{\theta_D}$ operating on $Z \in \mathbb{R}^{N \times d}$ and producing one logit per patch token. The discriminator is trained to distinguish clean teacher features $Z_{\text{ref}}$ (real) from corrected student features

$Z_{\text{corr}}$ (fake) using a BCE-based GAN objective:

$$\mathcal{L}_D = \frac{1}{2} \Big( \text{BCE}(D_{\theta_D}(Z_{\text{ref}}), 1) + \text{BCE}(D_{\theta_D}(Z_{\text{corr}}), 0) \Big),$$
$$\mathcal{L}_{\text{adv}} = \text{BCE}(D_{\theta_D}(Z_{\text{corr}}), 1). \tag{5}$$

By encouraging $Z_{\text{corr}}$ to be indistinguishable from clean features in distribution, $\mathcal{L}_{\text{adv}}$ constrains updates to stay on the clean feature manifold $\mathcal{M}$ and suppresses Euclidean shortcuts through low-density regions.

The adapter parameters are optimized by

$$\min_{\theta_A} \; \mathcal{L}_{\text{distill}} + \lambda \mathcal{L}_{\text{adv}}, \tag{6}$$

while the discriminator is optimized by $\min_{\theta_D} \mathcal{L}_D$. Here $\lambda$ controls the strength of the manifold alignment term.

### 3.5. Degradation Synthesis for Paired Training Data

Robustness adaptation requires paired data that captures realistic corruptions while remaining aligned with the vision encoder's pretraining distribution. Standard restoration datasets (*e.g.*, DIV2K (Agustsson & Timofte) and Flickr2K (Lim et al.)) provide clean references but lack domain diversity; adapting solely on them can induce feature drift from the VLM's operational manifold.

We therefore construct a multi-source hybrid dataset of approximately 75k paired samples by combining high-quality images (DIV2K, Flickr2K) with diverse web images from COCO (Lin et al.), LAION (Schuhmann et al., 2021), CC12M (Changpinyo et al., 2021), and TextVQA (Singh et al.). For each clean image, a degraded counterpart is synthesized using a high-order degradation pipeline inspired by Real-ESRGAN, covering blur, noise, ISP effects, and compression artifacts.

To reflect the spatial heterogeneity of real-world corruptions, we further apply semantic-aware local degradations. Specif-

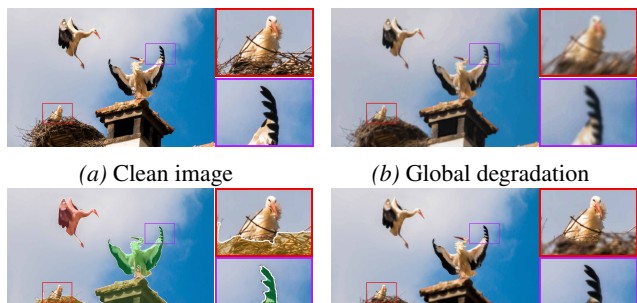

*(a)* Clean image      *(b)* Global degradation

*(c)* SAM2 mask overlay      *(d)* Local degradation

*Figure 6.* SAM2-based semantic-aware local corruption synthesis. We obtain semantic masks using SAM2 and apply degradations within randomly selected semantic regions, producing spatially heterogeneous corruptions that complement global degradations.

ically, we use SAM2 (Ravi et al., 2025) to sample object-level regions and apply degradations locally. As illustrated in Figure 6, this produces spatially heterogeneous artifacts where degradations are confined to semantic regions (*e.g.*, the bird) while the background remains clean, contrasting with uniform global degradations. Implementation details and synthesis parameters are provided in Appendix B.

## 4. Experiments

### 4.1. Experimental Setup

**Datasets and Benchmarks.** We evaluate robustness primarily on R-Bench (Li et al., 2025a), which assesses VLM performance under diverse corruption scenarios. R-Bench provides paired samples with a *Reference* (Ref) subset of high-quality images and a *Distraction* (Dis) subset containing degraded counterparts, spanning both synthetic corruptions and real-world environmental degradations. To verify that robustness adaptation does not compromise general multimodal capability, we additionally evaluate on MMBench, MMVet (Yu et al., 2024), LLaVABench, MathVista (Lu et al., 2024b), OCRBench (Liu et al., 2023b), and Hallusion-Bench (Guan et al., 2024). All evaluations are conducted using VLMEvalKit (Duan et al., 2024) with a fixed decoding configuration (temperature 0.2, max_new_tokens = 1024), and GPT-4.1 is used as the judge.

**Implementation Details.** We implement MAA on top of LLaVA-1.6-Mistral-7B with a CLIP ViT-L/14 vision encoder. Adapters are inserted into all $L = 24$ transformer layers of the vision encoder. MAA uses a hidden dimension of 256 and an attention dimension of 128, resulting in approximately 25.2M trainable parameters (under 10% of the ∼300M-parameter vision encoder), while the rest of the VLM remains frozen. Training is performed for one epoch with a global batch size of 32. We use AdamW with a learning rate of $5 \times 10^{-5}$ for adapters and $1 \times 10^{-6}$ for

the discriminator, and set the adversarial weight to $\lambda = 0.1$. All experiments are conducted on NVIDIA RTX PRO 6000 GPUs. Additional training and evaluation details are provided in Appendix D for reproducibility.

**Training Settings.** We freeze the VLM backbone and optimize only the adapter parameters $\theta_A$ and discriminator parameters $\theta_D$ using alternating GAN updates. For each paired sample $(x_{\text{ref}}, x_{\text{deg}})$, we first compute $Z_{\text{ref}}$ with the pristine teacher under no gradient, then compute $Z_{\text{corr}}$ with the adapter-augmented student. The discriminator $D_{\theta_D}$ is updated using $\mathcal{L}_D$ with gradients stopped on both branches, followed by updating the adapters using $\mathcal{L}_{\text{distill}} + \lambda \mathcal{L}_{\text{adv}}$, where gradients backpropagate through $D_{\theta_D}$.

**Inference Settings.** At inference time, the discriminator is discarded and only the learned adapters are retained within the frozen vision encoder. This results in a single-stage VLM with improved robustness and minimal overhead, avoiding the latency and brittleness of external two-stage restoration pipelines.

### 4.2. Robustness and Capability Preservation

We compare against four categories: (1) Base Model: the original LLaVA-1.6-Mistral-7B evaluated zero-shot. (2) Naive Adaptation (MSE-only): an MLP adapter trained only with feature-space MSE distillation. (3) External Restoration (Two-Stage): IR models prepended to the VLM, including Real-ESRGAN, AirNet, RAM-PromptIR (Qin et al., 2024), and MoCE-IR. (4) Representative Open-source VLMs: Other modern VLMs of similar parameter scales included for context (Lu et al., 2024a; Chen et al., 2024). For fair comparison, all adapter-based baselines are trained on the same paired data with the same optimization schedule. Implementation details are provided in Appendix D. All results are reported in Table 1 and we summarize the key findings below.

**Trade-offs of External Restoration.** Prepending strong IR models can yield mixed effects across benchmarks and rarely translates into consistent robustness gains. For example, Real-ESRGAN reduces MMVet accuracy from 42.33 to 38.12. We attribute this in part to distribution mismatch: IR models optimize perceptual fidelity (*e.g.*, PSNR/SSIM) and can introduce artifacts (*e.g.*, over-smoothing or ringing) that are atypical for web-scale pre-trained vision encoders. This may impair multimodal reasoning.

**Closing the Corruption-Induced Gap.** MAA achieves the best performance on R-Bench-Dis (59.39) and also improves Ref (59.92), indicating effective mitigation of corruption-induced representation shift. In contrast, Naive Adaptation (MSE-only) underperforms the base model on Dis (57.98 vs. 58.79), supporting the claim that paired MSE minimization alone is insufficient and can yield semantically misaligned representations without manifold constraints. We also ob-

*Table 1.* Evaluation results across various benchmarks. **Bold** denotes the best performance and underline denotes the second best performance. R-Bench is divided into Dis (Distraction) and Ref (Reference) subsets.

| Model | R-Bench | | MMBench | MMVet | LLaVABench | MathVista | OCRBench | HallusionBench |
| --- | --- | --- | --- | --- | --- | --- | --- | --- |
| | Dis | Ref | | | | | | |
| Base | 58.79 | 58.91 | 76.14 | 42.33 | 63.50 | 33.75 | 568 | 50.47 |
| Naive Adaptation | 57.98 | 58.30 | 74.63 | 42.02 | 62.00 | 33.90 | 562 | 50.36 |
| MAA (Ours) | **59.39** | **59.92** | **76.30** | **42.61** | 64.17 | **34.65** | 573 | **51.31** |
| +AirNet | 56.97 | 58.70 | 76.04 | 40.23 | 62.67 | 34.35 | **588** | 50.79 |
| +Real-ESRGAN | 57.17 | 58.70 | 75.45 | 38.12 | 63.67 | 34.25 | 567 | 50.26 |
| +RAM-PromptIR | 57.37 | 58.91 | 76.10 | 40.73 | 62.25 | 34.30 | 575 | 51.10 |
| +MoCE-IR | 58.38 | 59.31 | 76.00 | 40.96 | **64.83** | 33.70 | 576 | 50.79 |
| DeepSeek-VL | 54.95 | 55.87 | 63.80 | 29.20 | 51.10 | 30.70 | 413 | 27.60 |
| InternVL2-2B | 57.17 | 60.93 | 69.60 | 39.70 | 60.00 | 47.00 | 781 | 38.00 |

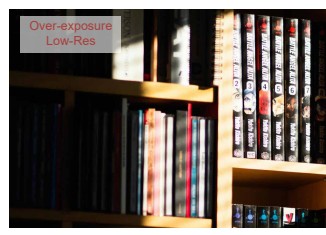 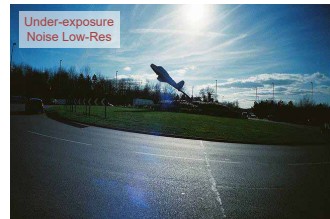 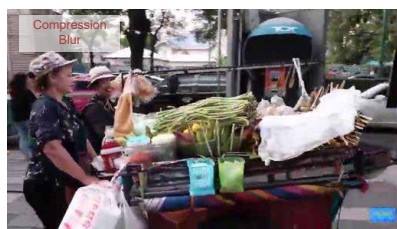

| | User | who was the author of the battle angel alta series? |
| --- | --- | --- |
| | LLaVA | **Yoko kishi** ✗ |
| | MAA (Ours) | **Yukito kishiro** ✓ |
| | Ground Truth | yukito kishiro |

| | User | Please describe this image in general |
| --- | --- | --- |
| | LLaVA | The image captures a moment of an airplane **in mid-flight**... ✗ |
| | MAA (Ours) | The image shows a **large airplane model** on a grassy area, positioned **as if it is taking off**... ✓ |
| | Ground Truth | A view of a plane that appears to be a statue. |

| | User | Which action is performed in this image? |
| --- | --- | --- |
| | Options | A. garbage collecting  B. pushing cart  C. celebrating  D. marching |
| | LLaVA | **A. garbage collecting** ✗ |
| | MAA (Ours) | **B. pushing cart** ✓ |
| | Ground Truth | B. pushing cart |

*Figure 7.* Qualitative comparison on real-world corrupted samples. We compare the base model (LLaVA-1.6) and MAA on three representative cases covering spatial and temporal corruptions. The base model is more prone to corruption-induced semantic errors (marked with ✗), whereas MAA yields more reliable responses (marked with ✓) by correcting corrupted representations within the VLM.

serve a modest gain on Ref, likely because web-sourced "Reference" images still contain mild intrinsic degradations (*e.g.*, sensor noise and compression) that are covered by our hybrid corruption synthesis (Appendix B).

**Capability Preservation on General Benchmarks.** MAA maintains comparable or slightly better performance on general multimodal benchmarks (*e.g.*, MMVet and Hallusion-Bench), suggesting that the adversarial manifold constraint helps preserve the semantics relied upon by the frozen LLM under robustness adaptation.

**MAA improves stability across clean and corrupted regimes.** MAA improves both Dis and Ref, suggesting that it enhances robustness without sacrificing performance on cleaner inputs. In contrast, external restoration acts as an input-space transform, and its effect depends on how the restored images match the vision encoder's pretraining distribution; this dependence can produce gains on some datasets or tasks but regressions on others, making transfer across clean and corrupted regimes less predictable.

*Table 2.* Efficiency comparison on R-Bench-Dis. Latency measures end-to-end inference time per image (ms). Our MAA achieves comparable performance with negligible overhead, whereas external IR methods incur a $5\times$ latency penalty.

| Method | GFLOPs | Latency | Speedup | Tokens |
| --- | --- | --- | --- | --- |
| Base (LLaVA-1.6) | 34,594 | 213 | $1.00\times$ | 4.91 |
| + Real-ESRGAN | 57,865 | 985 | $0.22\times$ | 4.87 |
| + AirNet | 43,067 | 1,024 | $0.21\times$ | 5.14 |
| + RAM-PromptIR | 38,599 | 1,081 | $0.20\times$ | 4.91 |
| + MoCE-IR | 63,956 | 1,013 | $0.21\times$ | 4.89 |
| **MAA (Ours)** | **34,765** | **230** | **0.93×** | 5.03 |

### 4.3. Efficiency Analysis

Table 2 compares inference efficiency on R-Bench-Dis. External restoration baselines require a two-stage pipeline (restoration followed by VLM inference), which substantially increases end-to-end latency, reaching roughly 1,000 ms per image in our setting. In contrast, MAA is integrated into the frozen vision encoder and introduces only a

*Table 3.* Ablation study on loss objectives and adapter architecture. **Bold** denotes the best result. Different components exhibit trade-offs between robustness on Dis and performance on Ref.

| Method | Components | | | R-Bench | |
|---|---|---|---|---|---|
| | Adv | W-Attn | Pool | Dis | Ref |
| Base (Zero-shot) | - | - | - | 58.79 | 58.91 |
| Naive Adaptation | × | × | × | 57.98 | 58.30 |
| MAA (MLP-Only) | ✓ | × | × | 59.19 | 60.12 |
| MAA w/o $\mathcal{L}_{adv}$ | × | ✓ | ✓ | 58.70 | 57.37 |
| MAA w/o W-Attn | ✓ | × | ✓ | 58.79 | **60.53** |
| MAA w/o Pooling | ✓ | ✓ | × | **59.39** | 59.11 |
| **MAA (Full)** | ✓ | ✓ | ✓ | **59.39** | 59.92 |

*Table 4.* Impact of different training data sources on robustness. A mixture of diverse domains yields the best generalization.

| Training Source | R-Bench | | $\Delta$ Dis |
|---|---|---|---|
| | Dis | Ref | |
| COCO | 56.57 | 58.91 | -2.22 |
| CC12M | 56.77 | 57.69 | -2.02 |
| TextVQA | 57.58 | 58.50 | -1.21 |
| Flickr2K | 58.18 | 58.70 | -0.61 |
| LAION | 58.18 | 58.91 | -0.61 |
| DIV2K | 58.99 | 59.11 | +0.20 |
| **Hybrid (Ours)** | **59.39** | **59.92** | **+0.60** |

*Table 5.* Sensitivity analysis of the adversarial weight $\lambda$ on R-Bench. We vary $\lambda$ while keeping the full MAA architecture fixed. $\lambda = 0$ removes only the adversarial manifold constraint. Positive $\lambda$ values consistently improve over the MSE-only variant.

| Metric | Adversarial Weight $\lambda$ | | | | | | |
|---|---|---|---|---|---|---|---|
| | 0 | 0.01 | 0.02 | 0.05 | 0.1 | 0.2 | 1.0 |
| R-Bench-Ref | 57.37 | 59.31 | 59.72 | 59.31 | **59.92** | 59.31 | **59.92** |
| R-Bench-Dis | 58.70 | 59.19 | 58.99 | 58.99 | **59.39** | 58.99 | 58.99 |

small overhead over the base model (230 ms vs. 213 ms, i.e., +17 ms), while keeping the generation length comparable (Tokens). Overall, MAA provides a favorable robustness-efficiency profile, avoiding the heavy latency cost of external restoration with near-base inference time.

### 4.4. Ablation Studies

**Impact of Adversarial Learning and Architecture.** Table 3 ablates both the training objective and the adapter design. The *Naive Adaptation* (trained without adversarial manifold alignment) underperforms the base model on R-Bench-Dis (57.98 vs. 58.79). In contrast, enabling the adversarial objective substantially improves robustness (*e.g.*, *MAA (MLP-Only)* reaches 59.19 on Dis), supporting our hypothesis that paired MSE distillation alone is insufficient and can yield semantically misaligned features without manifold constraints. Under adversarial training, removing Window Attention decreases R-Bench-Dis from 59.39 to 58.79, suggesting locality-aware token interactions are beneficial for handling corruptions. Pooling has a smaller effect on Dis in this setting (59.39 vs. 59.39) but changes the Dis/Ref trade-off, indicating interactions between robustness and clean-reference behavior.

**Impact of Training Data Strategy.** Table 4 evaluates models trained on subsets of our training data. Models trained on a single source generally underperform the proposed hybrid mixture on R-Bench-Dis, highlighting the importance of broad domain coverage. While high-quality datasets can provide strong clean anchors (*e.g.*, DIV2K reaches 58.99 on Dis), they are still inferior to the hybrid mixture (59.39 on Dis), and certain HQ-only settings (*e.g.*, Flickr2K) exhibit a noticeable gap, suggesting potential domain mismatch to the web-scale pre-training distribution. Overall, combining diverse web sources with high-quality references yields the best generalization.

**Sensitivity to the Adversarial Weight.** We further study the sensitivity of MAA to the adversarial weight $\lambda$ in Eq. 4, while keeping the adapter architecture, training data, and optimization schedule fixed. As shown in Table 5, removing

the adversarial term ($\lambda = 0$) leads to inferior performance, whereas introducing a positive adversarial weight consistently improves both R-Bench-Ref and R-Bench-Dis. The results are stable across a broad range of $\lambda \in [0.01, 1.0]$, indicating that MAA does not rely on delicate hyperparameter tuning. We use $\lambda = 0.1$ by default, which achieves the best R-Bench-Dis score and tied-best R-Bench-Ref score.

### 4.5. Attribution and Generalization Analysis

To disentangle the effect of adapter capacity from the proposed manifold-aware training objective, we further evaluate several adaptation mechanisms under the same paired training data. For each method, we compare its standard feature-distillation variant with a variant augmented by the adversarial manifold constraint. As shown in Table 6, plain feature matching is not sufficient: full fine-tuning severely damages the pretrained representation, and several non-invasive PEFT methods still underperform the zero-shot baseline without the adversarial term. In contrast, adding the adversarial manifold constraint consistently improves non-invasive adaptation methods and turns several drops into gains. MAA achieves the best performance on the corrupted Dis subset, suggesting that the robustness improvement is not merely due to adapter insertion or parameter count, but to the combination of conservative adaptation and manifold-aware regularization.

We also compare alternative feature-level objectives beyond MSE. Table 7 shows that distribution-aware objectives such as KL and MMD can already improve robustness, supporting our motivation that pointwise Euclidean alignment is

*Table 6.* Effect of the adversarial manifold constraint across adaptation mechanisms on R-Bench. "w/ Adv." adds the proposed adversarial manifold loss to the same adaptation mechanism. The zero-shot baseline is duplicated for comparison. Bold denotes the best performance and underline denotes the second best.

| Method | w/o Adv. | | w/ Adv. | |
|---|---|---|---|---|
| | Ref | Dis | Ref | Dis |
| Base (zero-shot) | **58.91** | **58.79** | 58.91 | 58.79 |
| Full FT | 33.60 | 32.93 | 34.41 | 34.34 |
| LoRA | 57.29 | 56.77 | 58.10 | 57.98 |
| Bottleneck MLP | 58.30 | 57.98 | **60.12** | 59.19 |
| Prefix Tuning (Jia et al., 2022) | 57.29 | 55.35 | **60.12** | 58.99 |
| MAA | 57.37 | 58.70 | 59.92 | **59.39** |

*Table 7.* Comparison of feature-level objectives on R-Bench. Distribution-aware objectives such as KL and MMD improve over plain feature matching, while the proposed MSE plus adversarial manifold objective achieves the best overall result. Bold denotes the best performance and underline denotes the second best.

| Loss / Objective | Ref | Dis |
|---|---|---|
| Base (zero-shot) | 58.91 | 58.79 |
| CORAL | 58.91 | 58.79 |
| InfoNCE | 0.40 | 0.00 |
| KL | 59.72 | **59.39** |
| MMD | 58.50 | 59.19 |
| MSE + Adv. (ours) | **59.92** | **59.39** |

insufficient. However, the proposed MSE plus adversarial manifold objective achieves the best Ref score and ties the best Dis score. By contrast, InfoNCE (van den Oord et al., 2018) collapses in this setting, likely because it optimizes relative similarity without anchoring features to the pretrained clean manifold, while CORAL (Sun et al., 2015) remains close to the baseline.

To further examine whether the proposed adaptation is tied to the LLaVA-1.6/CLIP pipeline, we evaluate MAA on InternVL3.5-30B-A3B, which uses a different VLM architecture and InternViT vision backbone. We use the same training configuration as in the main experiments without model-specific tuning.

### 4.6. Quantitative Analysis of Manifold Alignment

To empirically verify the SMG, we evaluate manifold alignment by quantifying the discrepancy between the repaired feature distribution and the clean reference distribution. We perform this evaluation on 2,000 paired images randomly sampled from the held-out split of our dataset (see Sec. 3.5), computing metrics in the projected patch-token space.

**Metrics.** We employ $MMD^2$, KID, and Energy Distance to measure distributional divergence, capturing differences in global statistics and higher-order moments. Additionally, we use CKA to quantify representational similarity,

*Table 8.* Cross-architecture evaluation on InternVL3.5-30B-A3B. MAA is applied with the same configuration as in the main LLaVA experiments, without model-specific tuning.

| Model / Adaptation | R-Bench-Ref | R-Bench-Dis |
|---|---|---|
| InternVL3.5-30B-A3B | 75.51 | 71.51 |
| + Naive Adaptation (MSE) | 71.26 | 68.89 |
| + MAA | **75.71** | **72.93** |
| Gain of MAA | +0.20 | +1.42 |

*Table 9.* Feature manifold alignment. Metrics are computed between the clean anchor $Z_{ref}$ and the features from: (1) Corrupted input (Base), (2) MSE-distilled adapter without manifold constraints (Naive Adaptation), and (3) MAA adapted input.

| Method | $MMD^2$ | KID | Energy | CKA |
|---|---|---|---|---|
| Base (Corrupted) | 0.0248 | 0.0062 | 0.2777 | 0.8252 |
| Naive Adaptation | 0.0506 | 0.0083 | 0.4823 | 0.8092 |
| **MAA (Ours)** | **0.0235** | **0.0059** | **0.2683** | **0.8467** |

assessing geometric consistency between feature sets.

**Key Observation (quantifying SMG).** As shown in Table 9, Naive Adaptation reduces the paired MSE objective by construction, yet it paradoxically increases distributional divergence ($MMD^2$/KID) and degrades representational similarity (CKA) compared to the corrupted baseline. This quantitative evidence confirms SMG: minimizing a pointwise Euclidean distance can drive features closer to targets individually, while pushing the population distribution away from the clean manifold.

**Effect of MAA.** MAA significantly improves alignment across all metrics. This indicates that the adversarial manifold constraint effectively restricts the corrected features to the in-distribution region, preserving the semantic structure required for downstream reasoning.

## 5. Conclusion

We study robustness degradation of VLMs under common real-world image corruptions and show that parameter-efficient adaptation that solely reduces feature MSE to clean references can be counterproductive. We address this issue with parameter-efficient adapters inserted into a frozen vision encoder and trained with a dual objective that couples paired feature distillation with an adversarial distribution alignment constraint. Across robustness and general capability benchmarks, the proposed method improves performance under corruptions while preserving overall multimodal capability, and incurs only a small overhead compared to two-stage external restoration. Future work will extend this framework to broader corruption distributions and video inputs, and explore more stable distribution-alignment objectives with weaker supervision.

## Acknowledgments

This work was supported in part by the National Natural Science Foundation of China under Grants 62476077 and 62272142.

## Impact Statement

This paper studies parameter-efficient adaptation to improve the robustness of vision-language models under common natural image corruptions (*e.g.*, blur, noise, compression, and illumination changes). Improved robustness may benefit real-world applications where image quality varies, by reducing performance degradation and potentially increasing the reliability of multimodal recognition and reasoning.

At the same time, robustness improvements do not imply guarantees of safety, fairness, or factuality: models may still exhibit hallucinations, biases, or other failure modes, and should be evaluated carefully before deployment, especially in high-stakes settings. In addition, more robust perception could be misused to enable model operation in adverse or covert capture conditions; responsible deployment should consider such risks. Our study does not introduce new data collection and does not rely on face recognition, surveillance scenarios, or sensitive personal data.

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

# A. Theoretical Analysis of Semantic Misalignment

This section provides a formal derivation demonstrating that the Mean Squared Error (MSE) objective inherently produces feature representations that deviate from the underlying data manifold. We prove that in high-dimensional spaces, the feature vector minimizing the MSE loss resides in a region of vanishing probability density relative to the true semantic manifold.

## A.1. Problem Formulation and Assumptions

Let $\mathcal{Z} \subseteq \mathbb{R}^d$ denote the feature space of the pre-trained vision encoder. Let $Z$ be a random variable representing the clean feature representation, and $\tilde{Z}$ denote the feature representation of the degraded input. We postulate two assumptions regarding the statistical structure of the semantic manifold and the nature of the degradation.

**Assumption 1 (Multimodal Mixture Distribution).** The distribution of clean features $P(z)$ is modeled as a mixture of $K$ isotropic Gaussian components, where each component corresponds to a distinct semantic category. The probability density function is given by:

$$P(z) = \sum_{k=1}^{K} \pi_k \mathcal{N}(z; \mu_k, \sigma^2 I_d), \tag{7}$$

where $\mu_k \in \mathbb{R}^d$ represents the prototype of the $k$-th category, $\pi_k$ is the prior probability, and $\sigma^2$ represents the intra-class variance. We assume the semantic categories are well-separated, satisfying $\|\mu_i - \mu_j\|_2 \gg \sigma$ for all $i \neq j$. The semantic manifold $\mathcal{M}$ is defined as the union of regions with high probability density, centered around $\{\mu_k\}_{k=1}^{K}$. We further assume that semantic prototypes are in general position, such that the expected squared Euclidean distance between two distinct class means scales linearly with the feature dimension $d$.

**Assumption 2 (Degradation-Induced Posterior Uncertainty).** The degradation process results in information loss such that the posterior distribution $P(z|\tilde{z})$ is not concentrated on a single mode. Specifically, we assume there exist at least two distinct categories $i$ and $j$ with non-zero posterior probabilities $w_i(\tilde{z})$ and $w_j(\tilde{z})$, where $w_k(\tilde{z}) \triangleq P(\text{Class} = k|\tilde{Z} = \tilde{z})$.

## A.2. Proof of Manifold Deviation

We now analyze the properties of the estimator derived from minimizing the MSE objective.

**Theorem A.1** (Vanishing Density of the MSE Estimator). *Let $\hat{z}_{MSE}$ be the optimal estimator minimizing the $L_2$ loss $\mathbb{E}[\|\hat{z} - z\|_2^2]$. Under Assumptions 1 and 2, $\hat{z}_{MSE}$ is a convex combination of the prototypes $\{\mu_k\}$. Furthermore, for a mixture of two equiprobable components separated by a distance $\Delta = \|\mu_i - \mu_j\|_2$, the probability density at the estimator $\hat{z}_{MSE}$ decays exponentially with respect to $\Delta^2$ compared to the density on the manifold.*

*Proof.* The global minimizer of the MSE objective is the conditional expectation of the target variable:

$$\hat{z}_{MSE} = \mathbb{E}[z|\tilde{z}] = \int_{\mathbb{R}^d} z P(z|\tilde{z}) dz. \tag{8}$$

Under Assumption 1, and approximating the class-conditional densities as Dirac deltas in the limit of small $\sigma$, the conditional expectation becomes:

$$\hat{z}_{MSE} = \sum_{k=1}^{K} w_k(\tilde{z}) \mu_k. \tag{9}$$

Since $\sum w_k(\tilde{z}) = 1$ and $w_k(\tilde{z}) \geq 0$, the vector $\hat{z}_{MSE}$ lies within the convex hull of the prototypes $\{\mu_k\}$. Given Assumption 2, $\hat{z}_{MSE}$ is strictly in the interior of the convex hull and does not coincide with any single prototype $\mu_k$.

We quantify the validity of this representation by comparing the probability density at $\hat{z}_{MSE}$ against the density at a manifold prototype $\mu_i$. To make the analysis concrete, we analyze a representative worst-case scenario with balanced posterior mass. Specifically, we consider the simplified case of binary uncertainty where $w_i(\tilde{z}) = w_j(\tilde{z}) = 0.5$.

The squared Euclidean distance from the midpoint to the nearest prototype $\mu_i$ is:

$$\|\hat{z}_{MSE} - \mu_i\|_2^2 = \left\| \frac{\mu_j - \mu_i}{2} \right\|_2^2 = \frac{\Delta^2}{4}. \tag{10}$$

The ratio of the probability density at the estimator $\hat{z}_{MSE}$ to the density on the manifold $\mu_i$ is:

$$\rho = \frac{P(\hat{z}_{MSE})}{P(\mu_i)} \approx \frac{\exp\left(-\frac{\|\hat{z}_{MSE}-\mu_i\|_2^2}{2\sigma^2}\right)}{\exp(0)} = \exp\left(-\frac{\Delta^2}{8\sigma^2}\right). \tag{11}$$

In high-dimensional feature spaces, the Euclidean distance between two randomly sampled vectors from distinct semantic classes typically scales with the square root of the dimension $d$. Therefore, $\Delta^2 \propto d$. Consequently, the density ratio satisfies:

$$\lim_{d\to\infty} \rho = 0. \tag{12}$$

$\square$

### A.3. Implication for Robustness

The derivation above proves that optimizing solely for Euclidean proximity drives the feature representation into a low-density region of the feature space. While $\hat{z}_{MSE}$ minimizes the geometric distance to the clean reference, it possesses a negligible probability of belonging to the data distribution defined by the pre-trained encoder. This mathematical result substantiates the existence of the SMG and justifies the necessity of the adversarial manifold constraint introduced in our proposed method.

## B. Data Synthesis and Construction Details

To ensure our MAA generalize to real-world corruptions while maintaining alignment with the pre-trained VLM distribution, we design a comprehensive data synthesis pipeline. This pipeline moves beyond simple Gaussian noise, incorporating a high-order degradation model and a novel semantic-aware local corruption strategy.

### B.1. Synthesis Philosophy and Multi-Source Composition

Standard image restoration datasets (*e.g.*, Flickr2K, DIV2K) offer high-quality references but lack the domain diversity of the web-scale data used to pre-train vision encoders like CLIP. Training solely on super-resolution datasets creates a domain gap, potentially leading to feature misalignment.

To bridge this gap, we construct a **Multi-Source Hybrid Dataset** consisting of 75,630 valid paired samples. Note that slight deviations from the planned counts (*e.g.*, 20k per subset) occur because the degradation pipeline performs strict validity checks, discarding images that are corrupted, possess non-standard channels (*e.g.*, CMYK/Grayscale), or fail the SAM 2 segmentation process. We curate images from five distinct domains to cover diverse semantic contents: high-quality photography, general web scenes, and text-heavy images. Table 10 details the composition of our training data.

*Table 10.* Detailed composition of the training dataset. The actual pair counts vary slightly due to strict quality filtering (*e.g.*, removing corrupt or unreadable source images during synthesis).

| Source Domain | Dataset | Versions/Img | Target | Valid Pairs |
|---|---|---|---|---|
| High-Quality (HQ) | Flickr2K | 3 | 11k | 7,436 |
| | DIV2K | 3 | 2.4k | 2,531 |
| General Web | COCO 2017 | 2 | 20k | 18,760 |
| Web-Scale Alignment | LAION Subset | 2 | 20k | 18,751 |
| Conceptual | CC12M Subset | 2 | 20k | 18,791 |
| OCR / Text | TextVQA | 2 | 10k | 9,361 |
| Total | - | - | $\sim$83k | 75,630 |

### B.2. High-Order Global Degradation Pipeline

We model the degradation process as a sequence of operations simulating physical acquisition and digital processing. Formally, a degraded image $x_{\text{deg}}$ is generated from a clean reference $x_{\text{ref}}$ via a chain of differentiable and non-differentiable operators:

$$x_{\text{deg}} = \mathcal{D}_{\text{compression}}(\mathcal{D}_{\text{digital}}(\mathcal{D}_{\text{sensor}}(\mathcal{D}_{\text{optics}}(x_{\text{ref}})))) \tag{13}$$

The pipeline consists of four stages, with parameters sampled stochastically from defined ranges (see Table 11):

**Stage 1: Optics & Motion (blur/resize).** We simulate optical blur using three kernels: (1) **Isotropic Gaussian**, (2) **Anisotropic Gaussian** (to model astigmatism), and (3) **Motion Blur** (linear kernels with random angles). We also apply random downsampling followed by upsampling using varying interpolations (area, bilinear, bicubic) to simulate low-resolution acquisition.

**Stage 2: Sensor Artifacts (noise).** We inject noise to simulate sensor imperfections:

- **Gaussian Noise:** Added in either RGB or grayscale space.

- **Band Noise:** We simulate readout circuitry artifacts by adding horizontal or vertical noise bands with random widths and intensities.

**Stage 3: Digital Image Processing (ISP).** To simulate the camera Image Signal Processor (ISP) and post-editing, we apply a randomized loop (1-2 iterations) of:

- **JPEG Compression:** Intermediate compression with variable quality tiers.

- **Color Quantization:** Reducing color bit-depth (*e.g.*, to 5-7 bits).

- **Color/Brightness Jitter:** Random shifts in saturation, luminance, and contrast.

**Stage 4: Output Rendering.** A final pass includes **Sinc filtering** to introduce ringing artifacts common in sharpened images, followed by a final JPEG compression stage to mimic web transmission.

### B.3. Semantic-Aware Local Degradation

Real-world degradations are rarely uniform; for instance, motion blur typically affects specific moving objects, while depth-of-field blur affects backgrounds. To model this spatial heterogeneity, we propose a **Semantic-Aware Local Degradation** strategy utilizing the Segment Anything Model 2 (SAM 2).

**Region Generation.** For a clean image $x$, we employ the **SAM 2 Automatic Mask Generator** to produce a set of candidate semantic masks. The generator is configured with a dense sampling grid (`points_per_side=32`) and strict quality filtering (`pred_iou_thresh=0.88`, `stability_score_thresh=0.95`). To ensure meaningful local degradations, we post-process the raw masks by decomposing connected components and filtering out small regions (area $< 1500$ pixels). Finally, we select a subset of up to 5 non-overlapping regions using a greedy strategy sorted by mask area to serve as targets for local corruption pipelines.

**Locally-Varying Synthesis.** We construct the final degraded image by blending globally degraded and locally degraded instances. Let $\mathcal{D}_g$ be the global degradation pipeline and $\mathcal{D}_l^{(i)}$ be a unique degradation pipeline sampled specifically for region $i$:

$$x_{\text{final}} = \left(1 - \sum_i \hat{m}_i\right) \odot \mathcal{D}_g(x) + \sum_i \hat{m}_i \odot \mathcal{D}_l^{(i)}(x) \tag{14}$$

where $\hat{m}_i$ is the mask $m_i$ softened by Gaussian feathering (radius=6) to prevent artificial edges at mask boundaries.

In our training set generation, we employ a probabilistic mixture strategy: **70%** of images undergo Hybrid degradation (Global + Local), **20%** undergo Global-only degradation, and **10%** undergo Local-only degradation (where the background remains clean).

### B.4. Hyperparameter Configurations

We generate two versions of degraded images for robustness training: *Mild* and *Severe*. The parameter ranges for these configurations are detailed in Table 11.

*Table 11.* Hyperparameter configurations for the degradation pipeline. "Prob" denotes the probability of applying an operator.

| Category | Parameter | Mild Setting | Severe Setting |
|---|---|---|---|
| Blur | Blur Probability | 0.90 | 0.95 |
| | Kernel Size Range | [7, 19] | [7, 21] |
| | Gaussian $\sigma$ | [0.2, 3.5] | [0.3, 5.0] |
| | Motion Blur Prob. | 0.20 | 0.30 |
| | Motion Kernel Size | [9, 21] | [11, 27] |
| Noise | Gaussian Noise Prob. | 0.80 | 0.85 |
| | Noise $\sigma$ (Base) | [1.0, 15.0] | [2.0, 20.0] |
| | Noise $\sigma$ (Strong) | [8.0, 25.0] | [10.0, 35.0] |
| | Band Noise Prob. | 0.20 | 0.30 |
| Digital (ISP) | Loop Iterations | [1, 2] | [1, 3] |
| | Color Quantization | [5, 7] bits | [4, 7] bits |
| | Saturation Factor | [0.8, 1.2] | [0.75, 1.25] |
| | Intermediate JPEG Quality | [25, 95] | [20, 95] |
| Output | Final JPEG Prob. | 0.80 | 0.85 |
| | Final JPEG Quality | [35, 95] | [30, 95] |
| | Sinc Filter Prob. | 0.25 | 0.35 |

# C. Additional Motivation Analysis

This section provides detailed experimental setups and quantitative justifications for the motivation studies presented in the main text (Figure 1 and Figure 2).

## C.1. Controlled Degradation Levels (L01-L10)

To rigorously analyze the correlation between feature deviation and semantic degradation (Figure 1), we synthesized 10 discrete degradation levels (L01-L10). These levels are derived by linearly interpolating the degradation parameters from a clean state (L00) to the upper bound of our "Severe" configuration (L10). Specifically, for a degradation parameter $p$ with a maximum severity $p_{\max}$, the value at level $k$ is given by $p_k = \frac{k}{10} \times p_{\max}$. The key parameter steps are detailed in Table 12.

*Table 12.* Complete parameter definitions for degradation levels L01-L10. Parameters are linearly interpolated from the mildest setting (L01) to the maximum severity (L10) used in our controlled motivation study.

| Parameter | L01 | L02 | L03 | L04 | L05 | L06 | L07 | L08 | L09 | L10 |
|---|---|---|---|---|---|---|---|---|---|---|
| Gaussian Noise ($\sigma$) | 3.5 | 7.0 | 10.5 | 14.0 | 17.5 | 21.0 | 24.5 | 28.0 | 31.5 | 35.0 |
| Blur Kernel Size | 3 | 5 | 7 | 9 | 11 | 13 | 15 | 17 | 19 | 21 |
| JPEG Quality | 95 | 88 | 80 | 73 | 65 | 58 | 50 | 43 | 36 | 30 |
| Color Jitter (Sat.) | 0.05 | 0.10 | 0.15 | 0.20 | 0.25 | 0.30 | 0.35 | 0.40 | 0.45 | 0.50 |

## C.2. Justification for Dataset Selection: Why Flickr2K?

In our controlled study, we utilized Flickr2K as the source of clean reference images. We justify this choice by conducting a comprehensive quality assessment across seven popular datasets. The goal is to identify a dataset that minimizes intrinsic degradations (*e.g.*, sensor noise, compression artifacts) that could confound our "clean vs. degraded" analysis.

We employed two evaluation protocols:

1. **LLM-based Assessment (Gemini 2.5 Flash):** We sampled ∼200 images from each dataset and prompted Gemini 2.5 Flash to score image quality on a scale of 0-100 and categorize them into five tiers (Perfect to Unusable).

2. **No-Reference IQA (MANIQA):** We used the MANIQA model (Yang et al., 2022) to predict perceptual quality scores.

**Results.** As shown in Table 13 and Table 14, standard pre-training datasets like COCO and even the reference split of R-Bench exhibit relatively low mean scores (∼65/100), with the majority of images classified merely as "Acceptable."

This confirms that these datasets suffer from significant intrinsic degradations. In contrast, super-resolution datasets like DIV2K and Flickr2K achieve significantly higher scores ($\sim$76-79/100) with a large proportion of "High Quality" images. We selected Flickr2K over DIV2K due to its larger scale and diverse content, providing a robust, high-quality baseline for controlled degradation synthesis.

*Table 13.* Dataset Quality Assessment via Gemini 2.5 Flash. Datasets are sorted by mean score. Flickr2K and DIV2K demonstrate superior quality consistency compared to standard web-crawled datasets like COCO.

| Dataset | Count | Mean | Median | Std | Quality Tier Distribution (Count) | | | | |
|---|---|---|---|---|---|---|---|---|---|
| | | | | | Perfect | High-Qual. | Acceptable | Low-Qual. | Unusable |
| DIV2K | 200 | 79.49 | 78.0 | 11.13 | 12 | 86 | 96 | 6 | 0 |
| HQ-50K (Yang et al., 2023) | 199 | 77.34 | 75.0 | 14.46 | 27 | 57 | 94 | 19 | 2 |
| **Flickr2K** | 200 | 76.19 | 72.0 | 10.87 | 11 | 60 | 120 | 9 | 0 |
| FoundIR-test-GT (Li et al., 2025b) | 122 | 72.34 | 70.0 | 10.39 | 2 | 27 | 86 | 7 | 0 |
| COCO | 200 | 65.93 | 68.0 | 11.28 | 1 | 19 | 134 | 45 | 1 |
| R-Bench-Ref | 202 | 62.29 | 65.0 | 10.95 | 2 | 5 | 127 | 67 | 1 |
| RealBlur-GT (Rim et al., 2020) | 233 | 55.61 | 55.0 | 11.02 | 0 | 2 | 93 | 129 | 9 |

*Table 14.* Dataset Quality Assessment via MANIQA. The trend is consistent with Gemini's assessment, confirming that Flickr2K provides a cleaner baseline than COCO or R-Bench-Ref.

| Dataset | Count | Mean | Median | Std | % Score $> 0.5$ | % Score $> 0.3$ |
|---|---|---|---|---|---|---|
| HQ-50K | 200 | 0.4394 | 0.4323 | 0.0916 | 31.50 | 7.00 |
| DIV2K | 200 | 0.4298 | 0.4214 | 0.0825 | 21.00 | 4.50 |
| **Flickr2K** | 200 | 0.4230 | 0.4152 | 0.0808 | 20.00 | 5.50 |
| COCO | 200 | 0.4075 | 0.4105 | 0.0894 | 14.00 | 9.50 |
| FoundIR-test-GT | 122 | 0.4044 | 0.4044 | 0.0698 | 9.84 | 5.74 |
| R-Bench-Ref | 202 | 0.3827 | 0.3746 | 0.0899 | 11.39 | 24.26 |
| RealBlur-GT | 234 | 0.3323 | 0.3294 | 0.0645 | 0.43 | 31.62 |

**In-House Benchmark Construction.** To measure semantic degradation (Figure 1b), we constructed a lightweight Visual Question Answering (VQA) benchmark using the selected Flickr2K images. We used Gemini 2.5 Pro to generate diverse questions and answers for each image based on its visual content. The generated QA pairs were then manually verified and cleaned by human annotators to ensure correctness and relevance, removing ambiguous or unanswerable questions. The metric reported is the Top-1 accuracy of the exact answer match.

### C.3. Training Dynamics Experiment Settings

Table 15 presents the exact numerical values corresponding to Figure 1(a) in the main text. It quantitatively demonstrates the trade-off between geometric alignment ($L_2$ distance) and semantic capability (MMBench Score).

*Table 15.* Quantitative analysis of training dynamics. Aggressively minimizing $L_2$ distance (*e.g.*, Full Fine-tuning) leads to severe degradation in semantic performance.

| Model Setting | Val $L_2$ Distance ($\downarrow$) | MMBench Score ($\uparrow$) |
|---|---|---|
| Base (Zero-shot) | 0.3222 | **76.14** |
| Adapter (MLP) | 0.1798 | 74.63 |
| Adapter (Transf.) | 0.1789 | 74.42 |
| LoRA ($r = 32$) | 0.1666 | 48.03 |
| Full Fine-tuning | **0.1560** | 45.16 |

For the training dynamics analysis (Figure 2 and Table 15), we compared four distinct PEFT configurations. All models were trained on the same subset of our synthesized paired data for 1 epoch with a batch size of 32, optimizing the simple MSE loss $\mathcal{L} = \|Z_{\text{deg}} - Z_{\text{ref}}\|^2$ without adversarial constraints.

- **Naive Adapter (MLP):** Identical to our MAA backbone (without adversarial training). Inserted into every layer. Hidden dimension $d_{hid} = 256$, $\sim$12.6M parameters.

- **Adapter (Transformer):** A heavier adapter design consisting of a full Transformer block inserted into each layer. Configuration: $L = 24$, $D = 1024$, bottleneck dimension $r = 128$, heads=4, FFN hidden size $4r = 512$. Parameter count per layer $\approx 458$k, total parameters $\approx 11.0$M.

- **LoRA:** Applied to the query and value matrices ($W_q, W_v$) of the vision encoder. Rank $r = 32$, alpha $\alpha = 64$, dropout $p = 0.05$.

- **Full Fine-tuning:** All parameters of the CLIP ViT-L/14 vision encoder ($\sim$304M) were unfrozen and updated.

Optimization was performed using AdamW with a learning rate of $5 \times 10^{-5}$.

## D. Implementation Details

### D.1. MAA Architecture and Training Hyperparameters

**Adapter Architecture.** We implement MAA on top of the LLaVA-1.6-Mistral-7B backbone, which utilizes a CLIP ViT-L/14 vision encoder (hidden dimension $D = 1024$). The MAA module is configured with the following specifications ("Medium" profile):

- **Token-wise MLP:** Input/Output dimension $D = 1024$, Hidden dimension $d_{mlp} = 256$.

- **Window Attention:** Attention dimension $d_{attn} = 128$, Number of heads $H = 8$, Window size $w = 7$.

- **Parameter Count:** Approximately 25.2M trainable parameters in total, constituting less than 10% of the vision encoder size.

**Discriminator Architecture.** The discriminator is a residual MLP operating on the projected token space (dimension $D_{llm} = 4096$). It consists of:

- An input projection layer ($4096 \rightarrow 2048$).

- A stack of **5 Residual Blocks**, where each block contains a two-layer MLP with LeakyReLU activation and LayerNorm. Hidden dimension $d_{hid} = 2048$.

- A final linear projection head ($2048 \rightarrow 1$) to output the real/fake logit per token.

**Training Configuration.** We train the adapters for 1 epoch on our paired dataset using the AdamW optimizer. Key hyperparameters include:

- **Global Batch Size:** 32.

- **Learning Rates:** LR$_G = 5 \times 10^{-5}$ for the Generator (Adapters); LR$_D = 1 \times 10^{-6}$ for the Discriminator.

- **Scheduler:** Cosine learning rate schedule with a 5% warmup ratio.

- **Adversarial Weight:** $\lambda = 0.1$.

- **Precision:** Mixed precision (FP16/BF16) training.

- **Hardware:** NVIDIA RTX PRO 6000 GPUs.

**Experimental Consistency.** To ensure fair comparison, all internal baselines (including the Naive Adapter and MLP-only variants) and ablation studies (structural components and data sources) are trained using the **identical hyperparameter configuration** (*e.g.*, learning rate, batch size, training steps) and hardware environment as the proposed MAA model, with the sole exception of the adversarial loss component being disabled where applicable.

*Table 16.* Multi-seed evaluation across benchmarks. Results are reported as mean $\pm$ standard deviation over seeds 0, 1, 2, and 42. MAA consistently improves over the baseline across robustness and general multimodal benchmarks.

| Method | R-Bench-Ref | R-Bench-Dis | MMBench | MMVet | LLaVABench | MathVista | HallusionBench | OCRBench |
|---|---|---|---|---|---|---|---|---|
| Baseline | $58.91 \pm 0.00$ | $58.64 \pm 0.05$ | $76.13 \pm 0.00$ | $40.09 \pm 0.96$ | $63.96 \pm 0.17$ | $33.66 \pm 0.06$ | $50.79 \pm 0.11$ | $578.50 \pm 3.52$ |
| MAA | $\mathbf{60.07 \pm 0.05}$ | $\mathbf{59.39 \pm 0.00}$ | $\mathbf{76.31 \pm 0.00}$ | $\mathbf{40.66 \pm 0.83}$ | $\mathbf{64.25 \pm 0.34}$ | $\mathbf{34.96 \pm 0.21}$ | $\mathbf{50.84 \pm 0.16}$ | $\mathbf{585.00 \pm 1.22}$ |

### D.2. Manifold Alignment Metrics Protocol

To quantitatively evaluate the Semantic Misalignment Gap, we compute distributional metrics on a hold-out set of $N = 2000$ paired samples. Feature representations are extracted from the output of the multimodal projector (excluding the `[CLS]` token) and averaged along the sequence dimension to obtain a global vector per image.

- **$MMD^2$ (Maximum Mean Discrepancy):** We use an unbiased estimator with a multi-scale RBF kernel. The bandwidths $\sigma$ are set to $\{1, 2, 4, 8\}$.

- **KID (Kernel Inception Distance):** Computed using a polynomial kernel with degree $d = 3$, $\gamma = 0.001$, and coefficient $c = 1.0$.

- **Kernel CKA:** We measure the Centered Kernel Alignment using the same RBF kernel mixture as MMD.

- **Energy Distance:** Computed using the unbiased estimator based on Euclidean distances between samples.

## E. Baseline Configurations

For all external image restoration baselines, we utilize the official pre-trained checkpoints provided by the respective authors. We employ these models as a frozen pre-processing stage before feeding the restored images into the LLaVA vision encoder.

- **AirNet:** We use the *All-In-One* checkpoint (`All.pth`, mode 3), which is trained jointly on denoising, deraining, and dehazing tasks.

- **Real-ESRGAN:** We use the `RealESRGAN_x4plus.pth` model, which provides a general-purpose restoration capability including super-resolution and artifact removal.

- **RAM:** We adopt the *Restore-Any-Model* framework equipped with the PromptIR backbone. Specifically, we use the fine-tuned checkpoint (`RAM-PromptIR-finetune`) optimized for handling diverse degradation types.

- **MoCE-IR:** We use the `MoCE_AIO5` checkpoint, which is trained to handle five corruption types: denoising, dehazing, deraining, deblurring, and low-light enhancement.

## F. Additional Stability Analysis

**Multi-seed Evaluation.** We evaluate the stability of MAA by rerunning the training with multiple random seeds, including seeds 0, 1, 2, and the main-paper seed 42. Table 16 reports the mean and standard deviation across these runs. MAA consistently improves over the baseline across robustness-oriented benchmarks and general multimodal benchmarks. In particular, the gains on R-Bench-Ref, R-Bench-Dis, MathVista, and OCRBench are stable across seeds, suggesting that the observed improvements are not caused by random initialization or evaluation variance.

## G. Extended Related Work

Due to space constraints, we provide a more detailed discussion here regarding adversarial robustness and the general application of Parameter-Efficient Fine-Tuning (PEFT), highlighting their distinctions from our work.

### G.1. Adversarial Robustness vs. Natural Robustness

Adversarial robustness has received significant attention in the VLM community. Benchmarks such as RobustBench (Croce et al., 2021) and OODRobustBench (Li et al., 2024) evaluate resilience to worst-case, imperceptible perturbations optimized to fool models. Prior work studies diverse attack paradigms, including multimodal attacks (Cui et al., 2024) and jailbreak-style threats (Luo et al., 2024a). Corresponding defenses range from robust training (Piras et al., 2025) to attribution-based analysis (Frosio & Kautz, 2023).

**Distinction.** It is important to distinguish these approaches from our focus. Adversarial perturbations are typically worst-case, artificial noise patterns, whereas we address *natural* image corruptions (*e.g.*, blur, sensor noise, compression) that arise from real-world acquisition processes. Techniques designed for adversarial defense often sacrifice clean accuracy significantly, which contradicts our goal of preserving general multimodal capabilities.

### G.2. Parameter-Efficient Fine-Tuning (PEFT)

PEFT methods, such as adapters and LoRA, have become standard for adapting Large Language Models (LLMs) and VLMs to new tasks or instructions (Zhu et al., 2024), enabling effective adaptation with minimal trainable parameters.

**Distinction.** While PEFT is widely effective for acquiring *new downstream skills* (*e.g.*, answering in a specific format), its utility for *robustness restoration* remains underexplored. As shown in our motivation study (Figure 2 in the main text), naively applying PEFT to minimize feature distance can induce representation drift—correcting the geometry while breaking the semantics. Our work repurposes the PEFT architecture but fundamentally changes the optimization objective (from task loss or simple MSE to manifold-adversarial alignment) to serve as an internal feature correction mechanism.

