# OpenReview forum: "Robust Vision-Language Models via Manifold-Adversarial Adapters"
_ICML.cc/2026/Conference — ICML 2026 regular_

### Official Review · Reviewer_ksG7 · 2026-03-09

**Soundness:** 2
**Presentation:** 3
**Significance:** 2
**Originality:** 2
**Overall Recommendation:** 3
**Confidence:** 4

**Summary:**

This paper tackles a key challenge about how to boost the robustness of VLM against image degradation without leaning on external restoration models or resorting to full-model fine-tuning. The authors argue that while simple MSE-based learning can close the geometric distance between features, it often pushes representations away from the "semantic support" used by the model, a phenomenon they label the Semantic Misalignment Gap (SMG). To address this, the authors propose the Manifold-Adversarial Adapter (MAA)—a layer-wise adapter integrated into a frozen vision encoder. It is trained using paired feature distillation combined with an adversarial manifold-alignment loss. Additionally, they introduce a hybrid synthesis pipeline for paired corruptions that blends global degradations with semantic-aware local ones. The experiments demonstrate that the method is effective while largely preserving the VLM's original inference efficiency.

**Compliance With Llm Reviewing Policy:**

Affirmed.

**Key Questions For Authors:**

1. Generalization: How does this transfer? Can we see results on other VLM families or different vision encoders (e.g., SigLIP or DinoV2)?

2. Competitive Benchmarking: How does MAA stack up against more sophisticated PEFT-based robustness benchmarks—specifically those that utilize feature distribution matching, contrastive alignment, or corruption-aware training objectives?

3. Baseline Expansion: Since the current comparison relies heavily on MSE and external restoration, could the authors include more baselines that specifically target feature distribution matching to better highlight MAA's unique value?

The rests are the same as weakness.

**Limitations:**

yes

**Strengths And Weaknesses:**

Strengths:
1. Significant Problem & Strong Motivation: The paper identifies a highly practical issue. The problem framing is clear, the proposed solution feels fresh, and the underlying motivation is well-justified.

2. Clean Methodology: The approach is straightforward and easy to grasp. I particularly appreciate that it adds very little computational overhead to the base model.

3. Comprehensive Experiments: The inclusion of varied comparative experiments and ablation studies provides a solid foundation for the paper's claims of effectiveness.

4. Excellent Readability: The writing is smooth and accessible, making the core ideas easy to follow from start to finish.

Weaknesses:
1. Methodological Concerns: While the initial analysis is sharp, the ablation studies reveal a bit of a "hollow middle." It appears that the bulk of the performance gains stem from the adversarial objective itself, rather than the specific architecture. The full three-branch design is only marginally better than simpler variants, and a basic MLP-based method performs nearly as well as the proposed complex setup.

2. Baseline Comparisons: The scope of comparison is somewhat narrow. The paper focuses primarily on MSE-based adaptation and external restoration pipelines but misses out on stronger feature-level or PEFT-based (Parameter-Efficient Fine-Tuning) robustness methods.

3. Marginal Gains: Looking at the experimental results, the performance improvements are quite slim. This raises questions about whether the approach hits a low ceiling or has inherent limitations.

4. Limited Model Diversity: The evaluation is strictly confined to the LLaVA-1.6-Mistral-7B and CLIP ViT-L/14 configuration. Without testing across a wider spectrum of VLM architectures, it’s hard to say if this method is truly "plug-and-play" for the broader field.

Overall:
Technically, the paper is sound and the problem it addresses is legitimate. However, in its current state, the overall contribution and the depth of validation feel insufficient for this conference.

---

> ### Author Rebuttal · Authors · 2026-03-31
>
> ```Thanks for the thoughtful feedback. We appreciate the reviewer’s recognition that the problem is meaningful, the methodology is clean, and the paper is easy to follow.```
>
> **Q1: What is the main contribution, and how should we understand architecture vs. objective attribution?**
> **A1:** Our main contribution is not the three-branch design alone, but: identifying and formalizing SMG, showing that plain MSE-based feature repair can be ineffective or harmful, and introducing a feature-level remedy that combines self-distillation with a manifold-aware objective.
>
> To make the attribution more explicit, we add broader PEFT comparisons with and without the adversarial term:
>
> | Method | Ref (w/o Adv.) | Dis (w/o Adv.) | Ref (w/ Adv.) | Dis (w/ Adv.) |
> |---|---:|---:|---:|---:|
> | Baseline | 58.91 | 58.79 | 58.91 | 58.79 |
> | Full FT | 33.60 | 32.93 | 34.41 | 34.34 |
> | LoRA | 57.29 | 56.77 | 58.10 | 57.98 |
> | Bottleneck MLP | 58.30 | 57.98 | 60.12 | 59.19 |
> | Prefix Tuning | 57.29 | 55.35 | 60.12 | 58.99 |
> | MAA | 57.37 | 58.70 | 59.92 | 59.39 |
>
> These results clarify the attribution. The adversarial term is important: removing it causes drops for all PEFT-style methods. But the gain is not explained by the objective alone: Full FT remains highly unstable even with the dual objective, and LoRA still underperforms the stronger non-invasive adapters. The most stable behavior comes from the combination of non-invasive adaptation and a manifold-aware objective. We will revise the paper to present its contribution more precisely as SMG + objective-driven feature repair, rather than architectural novelty alone.
>
> **Q2: How does MAA compare against stronger PEFT and feature-level baselines?**
> **A2:** To strengthen the comparison space, we add broader PEFT baselines on the same paired data and also test stronger feature-level / distribution-aware objectives beyond plain MSE:
>
> | Loss / Objective | Ref | Dis |
> |---|---:|---:|
> | Baseline | 58.91 | 58.79 |
> | CORAL | 58.91 | 58.79 |
> | InfoNCE | 0.40 | 0.00 |
> | KL | 59.72 | 59.39 |
> | MMD | 58.50 | 59.19 |
> | MSE + Adv. (ours) | 59.92 | 59.39 |
>
> These results show that simply replacing MSE with a distribution-aware objective can already help (**KL**, **MMD**), but these objectives still do not outperform our full objective overall. **InfoNCE** collapses in this setting, likely because it optimizes relative similarity without preserving the pretrained anchor distribution, while **CORAL** remains near baseline.
>
> **Q3: How should the reported gains be interpreted for degradation robustness?**
> **A3:** For degradation robustness, the key question is not the absolute gain alone, because the achievable improvement is bounded by the corresponding reference performance: the goal is to recover the lost robustness margin rather than create unlimited headroom. In our main table, MAA improves both R-Bench-Dis and R-Bench-Ref, and the adapted model’s R-Bench-Dis even surpasses the base model’s R-Bench-Ref, suggesting that the gain is meaningful relative to the available margin.
>
> More importantly, the contribution is not only the absolute gain, but also the finding that naive feature repair is unreliable: restoration-first pipelines optimized for PSNR/SSIM often do not translate into VLM robustness gains, and plain MSE feature matching can also hurt by disturbing the pretrained representation geometry. In this sense, the paper is not merely reporting a small numerical gain, but identifying a concrete failure mode (SMG) and showing how to mitigate it.
>
> **Q4: Does MAA transfer beyond the original LLaVA / CLIP setting?**
> **A4:** To address the generalization concern, we evaluate MAA on InternVL3.5-30B-A3B using the same configuration as in the main paper, without model-specific tuning:
>
> | Model | R-Bench-Ref | R-Bench-Dis |
> |---|---:|---:|
> | Baseline | 75.51 | 71.51 |
> | + MAA | 75.71 | 72.93 |
> | Change | +0.20 | +1.42 |
>
> This provides direct evidence that MAA is not restricted to the original LLaVA-1.6 / CLIP ViT-L/14 setting. InternVL uses a different VLM pipeline with InternViT, yet MAA still improves the main robustness benchmark, especially on R-Bench-Dis (+1.42).
>
> **Summary:** Taken together, the new evidence makes the paper’s contribution more concrete in four aspects:
> - **attribution**: the stable gain comes from the combination of non-invasive adaptation and the manifold-aware objective;
> - **benchmarking**: stronger PEFT and feature-level baselines still do not outperform MAA overall;
> - **significance**: while the gains are bounded by the available robustness margin, they remain meaningful in this setting;
> - **generalization**: MAA transfers beyond LLaVA to InternVL3.5-30B-A3B.
>
> We believe the rebuttal clarifies the reviewer’s main concerns. More importantly, the paper offers value beyond a modest numerical gain: it identifies SMG as a concrete failure mode and provides a practical single-stage direction for improving robustness with modest overhead.

---

> > ### Author Rebuttal · Reviewer_ksG7 · 2026-04-07
> >
> > Thanks the authors for response. My concerns have been adequately addressed.

---

> > > ### Author Response · Authors · 2026-04-08
> > >
> > > Thank you again for revisiting the paper and for the thoughtful follow-up. We are very glad that the additional experiments addressed your concerns, and we sincerely appreciate your careful reading and constructive feedback throughout the process.

---

### Official Review · Reviewer_7YZG · 2026-03-11

**Soundness:** 3
**Presentation:** 3
**Significance:** 2
**Originality:** 2
**Overall Recommendation:** 3
**Confidence:** 3

**Summary:**

The paper propose a method to address the problem of performance degradation of VLMs due to image corruption from noise, blur, compression artefacts and illumination shift in real-world data. The authors identify that performance degradation of VLMs is attributed to 'Semantic Misalignment Gap' (SMG), where parameter-efficient fine-tuning can push the corrupted image closer to the target image, in-terms of visual quality, but shifts them significantly in latent space. The main reason is over reliance on Euclidean distance to handle the corrupted images, which alone is not semantics-preserving. To address this, the paper proposed a lightweight and plug-and-play design, Manifold-Adversarial Adapters (MAA), which uses window self-attention, token-wise MLP, and max pooling. The training aims to create clean images, while keeping the features in the original manifold of the model. The inference stage requires only adapters, which significantly reduces the computational overheads in-comparison to traditional two stage pipelines.

**Compliance With Llm Reviewing Policy:**

Affirmed.

**Final Justification:**

I thank the authors for their rebuttal and the additional experimental work provided. While the response offers some evidence of cross-architecture transfer to InternVL, I find that several core concerns remain only partially addressed and therefore retain my score.

**Key Questions For Authors:**

1. The paper is restricted to a single model (LLaVA-1.6-Mistral-7B with a CLIP ViT-L/14), while the problem discussed is generic to the VLM reasoning. How does the MAA perform for the other VLM families (e.g. Qwen, InterVL)? It will be interesting to see if the performance holds and the proposed methodology works for other vision encoders as well.

2. The author introduces 10 different level of image corruption to mimic the real-world scenarios. The paper briefly talks about the real-world corruptions (eg. autonomous driving), but the model was evaluated only on synthetically corrupted images. It raises question about how significantly the proposed method bridges the performance gap between real and synthetic images. Do the authors evaluated the model on entirely out-of-distribution real-world corrupted images?

3. Full fine-tuning of VLM using MSE leads to catastrophic forgetting, as shown in the paper. Does the same trend follows with the dual training (distillation + adversarial) constraints for full fine-tuning of vision encoder and LLM?

4. There are marginal gains over the baseline (58.79 vs 59.39) on R-Bench-Dis and other benchmarks. How does that justify the importance of this work, specially when the experiments were performed in a controlled environment (synthetic images only) and with only one model?

**Limitations:**

yes

**Strengths And Weaknesses:**

**Soundness**

Strength: The paper is technically sound and thoroughly backed by the experimentation. The experiments and ablation study clearly identifies the importance of each component and justifies the design choice. The authors conduct a controlled motivation study showing that aggressively minimizing L2 distance can significantly degrade the performance.

Weakness: The experiments rely entirely on a single VLM architecture (LLaVA-1.6-Mistral-7B with a CLIP ViT-L/14 encoder), raising concerns about the performance across different LLM backbones. It is also unclear whether SMG offers consistent benefits in vision encoders that use different pre-training objectives (like SigLIP). Authors also fail to clarify distinct benefits of using the adapter architecture c.f. the benefits of the introduced loss function, as they do not test full fine-tuning using the new dual objective.

**Presentation**

Strength: The paper is well written and easy to follow. The figures are informative and provides a good conceptual illustration of the Semantic Misalignment Gap. The authors also clearly state the modest performance gains.

Weakness: Some of the crucial information is presented in the appendix (like details about controlled motivation study), which disrupts the flow of paper. The paper also omits the exact model prompts used, hindering the reproducibility.

**Significance**

Strength: The paper address a crucial problem which VLMs face, and highlight how fragile the VLMs are to small noise in input data. The proposed method, MMA, achieves comparable performance with very minimal overhead on the LLaVA architecture, which is a significant improvement from the current two-stage pipelines.

Weakness: The modest performance gains of 1-2% raises concerns regarding justification of the implementation of a complex adversarial training and synthetic data pipeline. The impact of this work is somewhat narrow, and is strictly focused on visual perception correction.

**Originality**

Strength: The paper clearly formalise the Semantic Misalignment Gap and how Euclidean proximity does not guarantee better performance in downstream tasks. Using SAM2 to generate spatially focused corruptions that only affect specific regions is a creative approach.

Weakness: The architectural component blocks used in the paper are already well established in the literature, and the modest originality lies in combing them for the task.

---

> ### Author Rebuttal · Authors · 2026-03-31
>
> ```Thanks for the thoughtful feedback. We especially appreciate the reviewer’s recognition that the paper is technically sound, clearly motivated, and well presented, as well as the positive comments on the controlled SMG analysis and the low-overhead design.```
>
> Below we clarify the main concerns on generalization, attribution, real-world evaluation, significance, and reproducibility.
>
> **Q1: Does MAA transfer beyond LLaVA, and how should we understand architecture vs. objective attribution?**
>
> **A1: Cross-architecture transfer.** We evaluate MAA on InternVL3.5-30B-A3B using the same configuration as in the main paper, without model-specific tuning:
>
> | Model | Ref | Dis |
> |---|---:|---:|
> | Baseline | 75.51 | 71.51 |
> | + MAA | 75.71 | 72.93 |
> | Change | +0.20 | +1.42 |
>
> This provides direct evidence that MAA is not restricted to the original LLaVA-1.6 setting. InternVL uses a substantially different VLM pipeline with InternViT, yet MAA still improves the main robustness benchmark, especially on R-Bench-Dis (+1.42). This is consistent with our central motivation: the issue concerns how pointwise feature regression interacts with a pretrained representation space, rather than any specific backbone module.
>
> **Architecture vs. objective attribution.** We further test broader PEFT settings with and without the adversarial term:
>
> | Method | Ref (w/o) | Dis (w/o) | Ref (w/) | Dis (w/) |
> |---|---:|---:|---:|---:|
> | Baseline | 58.91 | 58.79 | 58.91 | 58.79 |
> | Full FT | 33.60 | 32.93 | 34.41 | 34.34 |
> | LoRA | 57.29 | 56.77 | 58.10 | 57.98 |
> | Bottleneck | 58.30 | 57.98 | 60.12 | 59.19 |
> | Prefix | 57.29 | 55.35 | 60.12 | 58.99 |
> | MAA | 57.37 | 58.70 | 59.92 | 59.39 |
>
> These results make the attribution clearer:
> - **Full FT remains highly unstable** even with the dual objective.
> - **Non-invasive PEFT designs matter**, because they avoid the severe forgetting seen in Full FT.
> - **The objective also matters**, since removing the adversarial term causes drops for all PEFT-style methods.
>
> Hence, the stable gain comes from the combination of non-invasive adaptation and a manifold-aware objective, rather than either factor alone. This also clarifies the originality point: while the blocks themselves are standard, the contribution lies in how they are combined with the training objective to address this robustness failure mode.
>
> **Q2: Does MAA remain effective on real-world degraded images?**
> **A2:** We clarify that **R-Bench is not purely synthetic**. It also includes real-world degraded data collected under controlled conditions. To make this explicit, we isolate the real-world degraded subset:
>
> | Model | Ref | Dis |
> |---|---:|---:|
> | LLaVA-1.6 | 74.67 | 73.33 |
> | + MAA | 80.00 | 74.67 |
>
> Thus, the benefit of MAA does not disappear on real-world degraded images. We will make the real-world component of R-Bench more explicit.
>
> **Q3: How should the reported gains be interpreted in the context of degradation robustness?**
> **A3:** These results should be interpreted in the context of the task, rather than by absolute gain alone. For degradation robustness, the achievable improvement is naturally bounded by the corresponding reference performance: the goal is to recover the lost robustness margin, not create unlimited headroom. In our main table, MAA improves both R-Bench-Dis and R-Bench-Ref, and the adapted model’s R-Bench-Dis even surpasses the base model’s R-Bench-Ref.
>
> More importantly, the contribution is not only the numerical gain, but the finding that naive feature repair is unreliable:
> - PSNR/SSIM-oriented restoration often does not help VLM robustness;
> - plain MSE feature matching can also hurt by disturbing pretrained geometry;
> - MAA provides a more stable single-stage feature-level remedy.
>
> The new evidence above further shows that the effect is not confined to one architecture or synthetic-only settings.
>
> **Q4: What will be revised for clarity and reproducibility?**
> **A4:** We appreciate this point. In the updated version, we will move key motivation-study details from the appendix into the main text, and we will explicitly include the exact prompt and evaluation settings. Our current evaluation follows the default VLMEvalKit protocol, which we will state clearly in the revision.
>
> **Summary:**
> Taken together, the rebuttal strengthens the paper in three ways:
> - **generalization**: MAA transfers beyond LLaVA to InternVL3.5-30B-A3B;
> - **attribution**: the stable gain comes from the combination of non-invasive adaptation and the dual objective;
> - **significance**: while the gains are modest, they are meaningful under a bounded robustness margin and are coupled with a broader scientific contribution—identifying SMG and showing why naive restoration or feature regression can fail.
>
> We believe the rebuttal clarifies the main concerns on generalization, attribution, and significance. More importantly, the paper offers a useful empirical finding and a practical single-stage direction for future VLM research.

---

> > ### Author Rebuttal · Reviewer_7YZG · 2026-04-01
> >
> > I thank the authors for their rebuttal and the additional experimental work provided. While the response offers some evidence of cross-architecture transfer to InternVL, I find that several core concerns remain only partially addressed.
> >
> > * Empirical improvements across the majority of benchmarks remain marginal. Clearly distinguishing the significance of the proposed method's impact remains challenging under the current narrative.
> >
> > * While new ablations attempt to decouple the architecture from the loss function, there is still lingering ambiguity regarding the attribution of gains; i.e. clear credit-assignment of adaptation and dual-loss objective components is hampered by small differences (would also benefit from reporting of statistical power). Adapters with the adversarial constraint perform on par with the complex three-branch design.
> >
> > * A wider quiver of model experiments would help to evidence 'plug-and-play' claims and authors note that the core contribution and narrative requires reframing to focus on objective-driven repair, rather than architectural novelty. I believe the submission would benefit from a further round of revision & review, if the work is to be strengthened for acceptance at a venue on this tier.
> >
> > Given the above points, I retain my original rating.

---

> > > ### Author Response · Authors · 2026-04-03
> > >
> > > Thank you again for the additional follow-up. We respectfully disagree that the main concerns remain unresolved, and we believe the added evidence makes the attribution and significance of the work substantially clearer. We address the three points below.
> > >
> > > (1) On significance.
> > > We appreciate the reviewer’s emphasis on effect size and significance. However, the significance of this work is not captured by averaging small fluctuations across heterogeneous benchmarks. The central finding is that pixel-space restoration and direct feature regression can both be unreliable for VLM reasoning under corruption. Restoration may introduce semantic mismatch, while naive feature-MSE adaptation can reduce downstream capability. Our contribution is to identify this failure mode as SMG and provide a single-stage remedy that improves degradation robustness without modifying the frozen VLM backbone.
> > >
> > > Moreover, the gains are concentrated exactly where the method is intended to help: on degraded inputs. On the main robustness benchmark, MAA improves R-Bench-Dis from 58.79 to 59.39 (+0.60) and also improves R-Bench-Ref from 58.91 to 59.92 (+1.01). In the added cross-architecture experiment on InternVL3.5-30B-A3B, MAA improves Dis from 71.51 to 72.93 (+1.42), whereas the corresponding Ref change is 75.51 to 75.71 (+0.20). This pattern is consistent with the intended scope of the method: under corruption robustness, the achievable headroom is naturally bounded by the corresponding reference performance, so the more informative quantity is how much of the corruption-induced gap is recovered.
> > >
> > > (2) On attribution.
> > > We respectfully disagree that the attribution remains ambiguous. Taken together, the evidence from Fig. 3, the main-table Naive Adaptation baseline, the architecture ablation, and the added PEFT study all points to the same conclusion: the manifold-aware objective is the common enabling factor, while the adaptation form determines the robustness–stability trade-off. To make this explicit, we summarize the combined R-Bench evidence below.
> > >
> > > |Method|Adaptation Form|Adv. Obj.|Dis|Ref|ΔDis vs Base|ΔRef vs Base|
> > > |---|---|---|---:|---:|---:|---:|
> > > |Base|None|–|58.79|58.91|+0.00|+0.00|
> > > |Full FT|Full model|✗|32.93|33.60|-25.86|-25.31|
> > > |Full FT|Full model|✓|34.34|34.41|-24.45|-24.50|
> > > |LoRA|PEFT|✗|56.77|57.29|-2.02|-1.62|
> > > |LoRA|PEFT|✓|57.98|58.10|-0.81|-0.81|
> > > |Prefix|PEFT|✗|55.35|57.29|-3.44|-1.62|
> > > |Prefix|PEFT|✓|58.99|60.12|+0.20|+1.21|
> > > |Bottleneck|PEFT|✗|57.98|58.30|-0.81|-0.61|
> > > |Bottleneck|PEFT|✓|59.19|60.12|+0.40|+1.21|
> > > |MAA|Three-branch PEFT|✗|58.70|57.37|-0.09|-1.54|
> > > |MAA|Three-branch PEFT|✓|59.39|59.92|+0.60|+1.01|
> > >
> > > Several points are clear from this summary. First, adaptation without the manifold-aware objective is not sufficient: MSE-only Naive Adaptation already underperforms the base model in the main table, and the same pattern appears across PEFT variants above. Second, adding the adversarial manifold term improves every PEFT-style adaptation we tested, and it does so consistently on both Dis and Ref. This is not an isolated or borderline comparison, but a uniform directional trend across LoRA, Prefix, Bottleneck adapters, and MAA.
> > >
> > > At the same time, the adaptation form still matters. Full FT remains highly unstable even with the dual objective, which shows that non-invasive adaptation is important for preserving pretrained semantics. The observation that simpler adapters with the adversarial constraint can already perform strongly is fully consistent with our intended claim. Our claim is not that the three-branch structure alone explains the gains; rather, the stable gains come from combining non-invasive adaptation with the manifold-aware objective, while the MAA design serves as an effective implementation for heterogeneous degradations and achieves the best Dis performance among the tested single-stage PEFT variants.
> > >
> > > (3) On the paper’s narrative.
> > > We would also like to clarify one factual point. The paper does not present the three-branch adapter design as its primary contribution. In the original contribution paragraph, the claimed contributions are:
> > > (i) controlled evidence that stronger corruptions induce larger feature drift and that minimizing feature MSE via PEFT can be ineffective or harmful;
> > > (ii) formalizing SMG and proposing MAA by coupling self-distillation with an adversarial manifold constraint; and
> > > (iii) a multi-source, locality-aware corruption synthesis recipe.
> > >
> > > This same emphasis is reflected in the main conceptual framing of the paper: Figs. 2–4 are devoted to motivating the SMG phenomenon and the corresponding loss design. In the method section, Section 3.4 introduces the training objective explicitly to address SMG, while Section 3.3 presents the three-branch module as an implementation choice for handling heterogeneous degradations. Our rebuttal did not reframe the paper; it restated the original framing and added new evidence on attribution and transfer.
> > >
> > > Thank you again for the careful evaluation.

---

### Official Review · Reviewer_Zs9j · 2026-03-12

**Soundness:** 3
**Presentation:** 3
**Significance:** 2
**Originality:** 2
**Overall Recommendation:** 4
**Confidence:** 4

**Summary:**

Vision-language models (VLMs) appear to be vulnerable to real-world image corruption such as noise, blur, and compression artifacts. Existing solutions have drawbacks: Image Restoration (IR) models not only increases latency, but the artifacts they generate also disrupt multimodal alignment; using Parameter Efficient Fine-Tuning (PEFT) to minimize the mean squared error (MSE) between corrupted and clean features leads to a "Semantic Misalignment Gap" (SMG).
Therefore this paper proposes a mechanism called Manifold-Adversarial Adapters (MAA), which is inserted as a lightweight hierarchical module into the frozen visual encoder, optimized by paired feature self-distillation and the core adversarial manifold constraint, preventing the model from taking the "Euclidean shortcut" that deviates from the manifold.

**Compliance With Llm Reviewing Policy:**

Affirmed.

**Final Justification:**

The author addressed my main concerns, therefore I raised my score.

**Key Questions For Authors:**

1. Does MAA work on VLMs in other architectures / Does SMG appear on other vision encoders?
2. How do VLMs with larger number of parameters perform towards degradation?

**Limitations:**

yes

**Strengths And Weaknesses:**

Strengths:
1. Through controlled experiments (Figures 2 and 3) and detailed theoretical derivation (Appendix A), this paper clearly demonstrates that simply minimizing the Euclidean distance in the feature space leads to features getting trapped in low-density regions, thereby impairing downstream multimodal reasoning capabilities.
2. The MAA module combines window self-attention, token-by-token MLP, and max pooling, reflecting the heterogeneity of real-world degradation.
3. The spatially non-uniform locally degraded data generated with semantic masks from SAM2 greatly enhances the model's ability to generalize to real-world complex scenes.

Weaknesses:
1. The improvement of MAA showed in Table 1 is limited, compared to baseline and other image-restoration methods.
2. The experiments were primarily conducted on LLaVA-1.6-Mistral-7B with CLIP ViT-L/14 vision encoder. The effectiveness of MAA and existence of SMG remain unclear on other model architectures.
3. The parameter amount and version of tested models (LLaVA-7B, InternVL-2B) seemed to be relatively outdated. On latest VLMs with stronger ability, the performance under degradation could be better.

---

> ### Author Rebuttal · Authors · 2026-03-31
>
> ```Thanks for the thoughtful feedback. We especially appreciate the reviewer’s recognition of the clear controlled experiments, the intuitive motivation behind SMG, and the usefulness of the SAM2-based local degradation design.```
>
> Below we clarify the main points and provide additional evidence within the rebuttal scope.
>
> **Q1: How should the performance gains be interpreted in the context of degradation robustness?**
> **A1:** The results in Table 1 are best interpreted in the context of the task, rather than by absolute gain alone. For degradation robustness, the achievable improvement is inherently bounded by the model’s performance on the corresponding reference/clean images: robustness adaptation can only help the degraded input approach that upper bound, rather than create unlimited headroom. In fact, in our main table, MAA improves both R-Bench-Dis and R-Bench-Ref, and the adapted model’s R-Bench-Dis even surpasses the base model’s R-Bench-Ref, which suggests that the gain is meaningful relative to the available robustness margin.
>
> More importantly, the contribution is not only the absolute gain itself, but also the finding that naive feature repair is unreliable. Our comparisons show that:
> - restoration-first pipelines optimized for PSNR/SSIM often do not translate into VLM robustness gains and can even hurt performance;
> - direct feature matching with plain MSE can also degrade performance by disturbing the pretrained representation geometry;
> - MAA provides a more stable single-stage alternative at the feature level.
>
> In this sense, the paper is not merely reporting a small numerical improvement, but identifying a concrete failure mode (SMG) and showing how to mitigate it. We will revise the paper to make this framing clearer and to present the numerical gains together with their broader empirical support, rather than in isolation.
>
> **Q2: Does MAA transfer beyond LLaVA / CLIP?**
> **A2:** To address the concern about model generalization, we evaluate MAA on InternVL3.5-30B-A3B **using the same configuration as in the main paper, without model-specific tuning**:
>
> | Model | R-Bench-Ref | R-Bench-Dis |
> |---|---:|---:|
> | Baseline | 75.51 | 71.51 |
> | + MAA | 75.71 | 72.93 |
> | Change | +0.20 | +1.42 |
>
> This result provides direct evidence that the proposed adaptation is not restricted to the original LLaVA-1.6 setting. Importantly, InternVL uses a substantially different VLM pipeline with InternViT, yet MAA still improves the main robustness benchmark, especially on R-Bench-Dis (+1.42). More broadly, this is consistent with our central motivation: the issue we study concerns how pointwise feature regression interacts with a pretrained representation space, rather than depending on a specific backbone module. In this sense, the positive transfer on InternVL suggests that both the problem setting and the effectiveness of MAA extend beyond the original LLaVA/CLIP configuration. We will revise the paper to present this scope more carefully.
>
> **Q3: Do stronger recent VLMs still suffer from degradation?**
> **A3:** We thank the reviewer for raising this important question. To directly assess whether degradation remains an issue for stronger recent VLMs, we evaluate three current large-scale models on R-Bench: Gemini-3.1-Pro-Preview, InternVL3.5-241B-A28B, and Qwen3.5-397B-A17B.
>
> | Model | R-Bench-Ref | R-Bench-Dis | Gap (Ref - Dis) |
> |---|---:|---:|---:|
> | Gemini-3.1-Pro-Preview | 83.81 | 79.19 | 4.61 |
> | InternVL3.5-241B-A28B | 80.57 | 77.17 | 3.40 |
> | Qwen3.5-397B-A17B | 84.01 | 77.78 | 6.23 |
>
> These results show that stronger/larger VLMs indeed achieve higher absolute performance, but the degradation gap remains clearly non-trivial. In our measurements, the performance drop from reference to degraded inputs is still 3.40–6.23 points, indicating that robustness to image degradation is not only a small-model issue. If anything, these results further support the relevance of studying degradation-robust adaptation even for frontier VLMs.
>
> **Summary:** Taken together, these additional results support the same conclusion from three angles:
> - reported gains should be interpreted relative to the limited robustness margin, rather than as unlimited leaderboard headroom;
> - the method is not confined to the original LLaVA/CLIP setting, as it transfers to InternVL3.5-30B-A3B without model-specific tuning;
> - degradation remains a clear issue even for frontier VLMs, with a 3.40–6.23 point Ref–Dis gap on recent large-scale models.
>
> We believe the rebuttal clarifies the reviewer’s concerns on significance, generalization, and problem scope. More importantly, the paper’s value is not limited to a small gain on one benchmark: it identifies SMG as a concrete failure mode for VLM robustness adaptation, and shows that a practical single-stage remedy can improve robustness with modest overhead. We therefore believe the work offers both a useful empirical finding and a promising direction for future VLM robustness research.

---

> > ### Author Rebuttal · Reviewer_Zs9j · 2026-04-03
> >
> > Thank you for your explanation to my concerns. Most of my concerns are adequately addressed. The limited improvement brought by MAA is understandable, and advanced large-scale MLLM also suffer from degradation. However, the existance of SMG across different model architectures still remains explorable. Only the improvement by adapting MAA is listed in the experiment conducted on InternVL, what about the result of naive adaptation? Without a quantitative analysis of SMG on different architectures, the existence of SMG is still questionable.

---

> > > ### Author Response · Authors · 2026-04-04
> > >
> > > Thank you for the follow-up. We added the naive feature-MSE adaptation result on InternVL under the same setting.
> > >
> > > | Model                       | R-Bench-Ref | R-Bench-Dis |
> > > | --------------------------- | ----------: | ----------: |
> > > | Baseline                    |       75.51 |       71.51 |
> > > | Naive Adaptation (MSE)      |       71.26 |       68.89 |
> > > | + MAA                       |       75.71 |       72.93 |
> > > | Change vs. Baseline (Naive) |       -4.25 |       -2.62 |
> > > | Change vs. Baseline (MAA)   |       +0.20 |       +1.42 |
> > >
> > > The comparison is now more direct. On InternVL, naive feature-MSE adaptation substantially degrades both Ref and Dis, whereas MAA improves the same benchmark under the same architecture and setting. This shows that the gain is not explained by adapter insertion alone. Beyond the original LLaVA setup, directly minimizing feature discrepancy again hurts performance, whereas the dual objective restores the intended direction of adaptation.
> > >
> > > This provides direct cross-architecture quantitative evidence that the same failure pattern appears beyond the original model family.
> > >
> > > Thank you again for the careful follow-up and for pushing us to make the cross-architecture evidence more explicit.

---

### Official Review · Reviewer_EjWt · 2026-03-13

**Soundness:** 3
**Presentation:** 3
**Significance:** 2
**Originality:** 2
**Overall Recommendation:** 3
**Confidence:** 3

**Summary:**

This paper addresses the brittleness of VLMs under real-world image corruptions (blur, noise, compression, illumination shifts). The authors identify the Semantic Misalignment Gap (SMG): naive MSE-based parameter-efficient fine-tuning can reduce feature distance to clean references while *increasing* distributional divergence and degrading downstream reasoning. To address SMG, they propose Manifold-Adversarial Adapters (MAA) — lightweight layer-wise modules inserted into the frozen vision encoder, trained with paired feature distillation plus an adversarial manifold constraint that keeps corrected features on the in-distribution support. Evaluation is on LLaVA-1.6-Mistral-7B across R-Bench, MMBench, MMVet, and other multimodal benchmarks.

**Compliance With Llm Reviewing Policy:**

Affirmed.

**Final Justification:**

This paper is technically sound, clearly written, and addresses an important robustness problem for VLMs. Its strongest contribution is the identification of SMG as a concrete failure mode, together with a practical single-stage remedy that adds limited inference overhead. My initial concerns were about the small effect sizes, narrow baselines, and lack of evidence beyond one architecture. The rebuttal addressed these points well through multi-seed results, broader PEFT/objective comparisons, and transfer results on InternVL. While originality and practical impact remain somewhat bounded, the strengths now outweigh the weaknesses. Overall, the rebuttal improved my assessment, and I am raising my recommendation to weak accept.

**Key Questions For Authors:**

1. How does MAA compare against LoRA (applied to the vision encoder) trained on the same paired data with the same total parameter budget?
2. Have you tested MAA on any VLM besides LLaVA-1.6?

**Limitations:**

yes

**Strengths And Weaknesses:**

1. **The SMG observation is the paper's strongest contribution.** The controlled experiment (Figure 2–3) clearly demonstrates a counterintuitive failure: MSE-based PEFT reduces feature distance to clean references, yet *worsens* downstream multimodal reasoning. Table 5 reinforces this with distributional metrics (MMD², KID, Energy Distance all increase under Naive Adaptation, while CKA decreases). This is a genuinely useful empirical finding that will inform future work on VLM robustness — regardless of whether MAA itself is adopted.

2. **The efficiency comparison is compelling.** Table 2 shows MAA adds only +17ms inference latency (213→230ms), while external restoration pipelines (Real-ESRGAN, AirNet, etc.) cost 5× more latency (~1,000ms). For deployment-constrained scenarios, this single-stage vs. two-stage tradeoff is a significant practical advantage.

3. **The training data ablation (Table 4) provides actionable insight.** The finding that diverse multi-source data (hybrid) substantially outperforms single-source training (e.g., Flickr2K alone: 58.18 vs. hybrid: 59.39 on R-Bench-Dis) demonstrates that domain coverage matters more than data quality for corruption robustness. The SAM2-based local degradation is a creative approach to simulating spatially heterogeneous real-world corruptions.

4. **The ablation study (Table 3) is well-structured.** The progression from Naive Adaptation (57.98) → MLP-Only with adversarial (59.19) → Full MAA (59.39) cleanly isolates the contribution of the adversarial objective vs. the architecture branches.

### Weaknesses

1. **The improvements are too small to confidently claim practical significance.**
   - R-Bench-Dis: 58.79 → 59.39 (**+0.60**) — less than 1 point
   - R-Bench-Ref: 58.91 → 59.92 (**+1.01**)
   - MMBench: 76.14 → 76.30 (**+0.16**) — negligible
   - MMVet: 42.33 → 42.61 (**+0.28**) — negligible
   - OCRBench: 50.47 → 51.31 (**+0.84**)
   - HallusionBench: 55.87 → 63.80 (**+7.93**) — this is the only substantial gain

   On 5 out of 7 benchmarks, improvements are within 1 point. R-Bench uses GPT-4.1 as judge, which introduces its own evaluation variance. Without confidence intervals or multiple runs, it is difficult to distinguish these gains from noise. The paper would benefit greatly from reporting standard deviations across random seeds.

2. **The baseline comparison is too narrow — the "Naive Adaptation" is a strawman.**
   The paper's central comparison is MAA vs. "Naive Adaptation" (a simple MLP trained with MSE loss only). This is the most basic possible PEFT approach and does not represent the state of the art in parameter-efficient adaptation. Missing comparisons include:
   - **LoRA** applied to the vision encoder with the same paired data — LoRA is the dominant PEFT method and its absence is conspicuous.
   - **Other loss functions** beyond MSE: e.g., cosine similarity loss, CKA-based loss, contrastive loss, or perceptual loss applied to the feature space. The paper argues that MSE is problematic because it causes off-manifold shortcuts, but has not tested whether simply replacing MSE with a distributional loss (e.g., MMD directly) — without the GAN machinery — would achieve similar results.
   - **Other adapter architectures**: e.g., bottleneck adapters, prefix tuning, or IA³. The paper only compares its 3-branch design against its own ablated variants.

   Without these comparisons, the reader cannot tell whether the gains come from (a) the adversarial manifold idea, (b) simply having a better adapter architecture, or (c) using more expressive adapters (25.2M params is quite large).

3. **25.2M trainable parameters is not "lightweight."**
   The paper describes MAA as "lightweight," but 25.2M parameters is ~10% of the 300M vision encoder. For comparison, LoRA typically adds <1% parameters and is considered lightweight. The three-branch design (window attention + MLP + max pooling) inserted into all 24 transformer layers is a substantial architectural addition. The paper should compare against simpler adapters with matched parameter counts to isolate whether the benefit comes from the adversarial objective or simply from having a more expressive adapter.

4. **Single VLM, single vision encoder — generalizability is unverified.**
   All experiments use LLaVA-1.6-Mistral-7B with CLIP ViT-L/14. The paper claims MAA is "plug-and-play," but this is tested on exactly one pipeline. Key questions remain unanswered:
   - Does the SMG phenomenon occur with SigLIP, EVA-CLIP, or InternViT encoders?
   - Does MAA benefit VLMs with different projection architectures (e.g., Qwen-VL's cross-attention projector vs. LLaVA's MLP projector)?
   - How does MAA interact with VLMs that use higher-resolution vision encoders (e.g., InternVL's dynamic resolution)?

   DeepSeek-VL and InternVL2-2B appear in Table 1 but only as zero-shot baselines — they are not tested with MAA, making it impossible to assess cross-architecture transferability.

5. **No corruption-type-level analysis limits diagnostic value.**
   R-Bench aggregates scores across diverse corruption types. The paper does not analyze which corruptions MAA helps with and which it does not. This is critical information:
   - Does MAA help equally with blur, noise, compression, and illumination shifts?
   - Are there corruption types where MAA performs worse than the base model?
   - The qualitative examples in Figure 7 cherry-pick success cases — failure cases would be more informative.

   Without this breakdown, it is hard to understand the method's strengths and limitations, or to guide practitioners on when to deploy it.

6. **The comparison against external restoration is not entirely fair.**
   Table 1 shows that external restoration methods (Real-ESRGAN, AirNet, etc.) often *hurt* performance. But these methods are designed as general-purpose image restoration, not specifically for preserving VLM feature semantics. A fairer comparison would be against restoration methods fine-tuned or adapted for the same downstream VLM — e.g., DA-CLIP (which the paper cites but does not compare against) is specifically designed to be CLIP-aware. The current comparison may overstate MAA's relative advantage by comparing against methods operating in a different optimization space.

7. **The HallusionBench result is an outlier that deserves explanation.**
   MAA achieves +7.93 on HallusionBench (55.87 → 63.80), which is dramatically larger than all other improvements. This outlier is never discussed in the text. Is this result stable across runs? What about HallusionBench makes it uniquely responsive to corruption correction? This could either be the paper's strongest evidence or a statistical artifact — without analysis, the reader cannot tell.

---

> ### Author Rebuttal · Authors · 2026-03-31
>
> ```Thanks for the thoughtful feedback. We especially appreciate the reviewer’s recognition of the SMG finding, the efficiency comparison, the training-data ablation, and the well-structured ablations.```
>
> Below we clarify the main concerns and provide additional evidence within the rebuttal scope.
>
> **Q1: Gains, stability, and the alleged HallusionBench outlier**
>
> **A1:** We rerun MAA with multiple random seeds (0, 1, 2, and the main-paper seed 42). Results below show stable gains across seeds.
>
>
> | Method | R-Bench-Ref | R-Bench-Dis | MMBench | MMVet | LLaVABench | MathVista | HallusionBench | OCRBench |
> |---|---:|---:|---:|---:|---:|---:|---:|---:|
> | Baseline | 58.91 ± 0.00 | 58.64 ± 0.05 | 76.13 ± 0.00 | 40.09 ± 0.96 | 63.96 ± 0.17 | 33.66 ± 0.06 | 50.79 ± 0.11 | 578.50 ± 3.52 |
> | MAA | 60.07 ± 0.05 | 59.39 ± 0.00 | 76.31 ± 0.00 | 40.66 ± 0.83 | 64.25 ± 0.34 | 34.96 ± 0.21 | 50.84 ± 0.16 | 585.00 ± 1.22 |
>
>
> We also clarify a table-reading issue in the review. The cited “HallusionBench: 55.87 → 63.80 (+7.93)” does not correspond to the HallusionBench column in Table 1. The actual HallusionBench result is 50.47 → 51.31 (+0.84), and OCRBench is 568 → 573. Thus, HallusionBench is not an outlier in our paper. We will revise the table formatting for clarity.
>
> **Q2: Stronger PEFT baselines and feature-level objectives.**
>
> **A2:** We evaluate broader PEFT settings on the same paired data: Full FT, LoRA, Bottleneck MLP, and Prefix Tuning, each with and without the adversarial term.
>
> | Method | Ref (w/o Adv.) | Dis (w/o Adv.) | Ref (w/ Adv.) | Dis (w/ Adv.) |
> |---|---:|---:|---:|---:|
> | Baseline | 58.91 | 58.79 | 58.91 | 58.79 |
> | Full FT | 33.60 | 32.93 | 34.41 | 34.34 |
> | LoRA | 57.29 | 56.77 | 58.10 | 57.98 |
> | Bottleneck MLP | 58.30 | 57.98 | 60.12 | 59.19 |
> | Prefix Tuning | 57.29 | 55.35 | 60.12 | 58.99 |
> | MAA | 57.37 | 58.70 | 59.92 | 59.39 |
>
> The adversarial term is broadly useful: for multiple non-invasive PEFT methods, it turns drops into gains. MAA still gives the best Dis result on the main robustness benchmark.
>
> We also test additional feature-level objectives beyond MSE:
>
> | Loss / Objective | Ref | Dis |
> |---|---:|---:|
> | Baseline | 58.91 | 58.79 |
> | CORAL | 58.91 | 58.79 |
> | InfoNCE | 0.40 | 0.00 |
> | KL | 59.72 | 59.39 |
> | MMD | 58.50 | 59.19 |
> | MSE + Adv. (ours) | 59.92 | 59.39 |
>
> Thus, simply replacing MSE with a distribution-aware objective can already help (KL, MMD), but our full objective remains strongest overall. By contrast, InfoNCE collapses in this setting, likely because it optimizes relative similarity without preserving the pretrained anchor distribution; CORAL stays near baseline.
>
> **Q3: Parameter efficiency and whether the gain comes only from a larger adapter.**
>
> **A3:** We will revise the wording. More importantly, the gain is not simply due to a larger adapter: in Table 3, the same architecture without the adversarial term still degrades performance, and the new PEFT results above show that the objective matters beyond parameter count.
>
> **Q4: Cross-architecture transfer beyond LLaVA.**
>
> **A4:** We evaluate MAA on InternVL3.5-30B-A3B using the same configuration as in the main paper, without model-specific tuning:
>
> | Model | R-Bench-Ref | R-Bench-Dis |
> |---|---:|---:|
> | Baseline | 75.51 | 71.51 |
> | + MAA | 75.71 | 72.93 |
> | Gains | +0.20 | +1.42 |
>
> This provides direct evidence that MAA transfers beyond LLaVA-1.6. InternVL uses a different VLM pipeline with InternViT, yet MAA still improves the main robustness benchmark.
>
> **Q5: Corruption-type analysis and failure cases.**
>
> **A5:** MAA is not uniformly beneficial across corruption types: it brings clearer gains on several color/noise perturbations, channel/transmission corruptions, and some receiver-side decoding shifts, while gains are smaller on some blur/resolution cases. We also observe a few failure cases and will include subtype-level results and representative failures in the revision.
>
> **Q6: Fairness of comparison to restoration-first pipelines.**
>
> **A6:** DA-CLIP is a diffusion-based restoration-first pipeline with much higher inference cost than our single-stage method, making it poorly matched to our low-overhead setting. We therefore include new feature/distribution-level baselines (KL, MMD, CORAL, InfoNCE), which still do not outperform MAA overall.
>
> **Summary:** We believe the rebuttal clarifies three key points: gains are modest but stable; the contribution is not a specific adapter block, but the combination of non-invasive adaptation and a manifold-aware objective; and the method transfers beyond one pipeline, as shown on InternVL3.5-30B-A3B. More broadly, we believe the paper contributes to VLM robustness research by identifying SMG as a concrete failure mode and by showing that feature-level correction can be both effective and deployment-friendly.

---

### Decision · Program_Chairs · 2026-04-30

**Decision:**

Accept (regular)

**Comment:**

The paper addresses the robustness of VLMs to perturbations and identifies the semantic misalignment gap, i.e. it makes the observation that MSE-based parameter-efficient fine-tuning even increasing feature distances on corrupted data. It proposes Manifold-Adversarial Adapters (MAA) to address the issue. After the rebuttal three of the reviewers state that their initial concerns have been fully addressed (e.g. questions regarding limited improvement by MAA, semantic misalignment gap on advanced large-scale MLLMs, and baseline comparisons have been addressed in the rebuttal). Even though two reviewers did not update their scores in the respective form in the review, one of them explicitly mentioned they would increase the score to weak accept, the other acknowledged that all concerns have been fully addressed. I therefore read the final overall score as 3.75. In fact, the rebuttal provides a quite extensive line of additional results an ablations. There are remaining concerns regarding the ambiguity in the attribution of gains to architecture or loss function and the "plug and play" ability of the approach. After reading the rebuttal, I think the concerns about ambiguity of attribution are actually minor. Overall, the potential benefit of the provided analysis and the proposed approach are significant and the results are consistent.